# Biological Characterization and Evaluation of the Therapeutic Value of *Vibrio* Phages 4141 and MJW Isolated from Clinical and Sewage Water Samples of Kolkata

**DOI:** 10.3390/v16111741

**Published:** 2024-11-06

**Authors:** Sanjoy Biswas, Devendra Nath Tewari, Alok Kumar Chakrabarti, Shanta Dutta

**Affiliations:** ICMR-National Institute for Research in Bacterial Infections (Formerly “ICMR-National Institute of Cholera and Enteric Diseases”), P33, CIT Road, Scheme XM Beliaghata, Kolkata 700010, India; biswassanjoy1995@gmail.com (S.B.); devendranath15ti@gmail.com (D.N.T.); shanta.niced@icmr.gov.in (S.D.)

**Keywords:** bacteriophage, phage therapy, genomic analysis, antimicrobial resistant (AMR), biofilm degradation, *Vibrio cholerae*

## Abstract

The growing prevalence of antimicrobial resistance (AMR) necessitates the development of new treatment methods to combat diseases like cholera. Lytic bacteriophages are viruses that specifically target and lyse bacteria upon infection, making them a possible treatment option for multi-drug-resistant pathogens. The current study investigated the potential role of bacteriophages isolated from clinical stool and sewage water samples in treating multi-drug-resistant *Vibrio cholerae* infection, finding that over 95% of the strains were susceptible. Whole-genome sequencing (WGS) analysis revealed that both Vibrio phage 4141 (4141) and Vibrio phage MJW (MJW) contain double-stranded DNA genomes consisting of 38,498 bp (43% GC) and 49,880 bp (42.5% GC) with 46 and 64 open reading frames (ORFs), respectively. Transmission electron microscope (TEM) and WGS analysis of Vibrio phage 4141 and Vibrio phage MJW validated that they are classified under the family *Autographiviridae* and *Zobellviridae,* respectively. Furthermore, both the phages showed highly significant biofilm degradation properties. The characterization of the phages and their strict host range, high spectrum of lytic ability, high efficiency of biofilm degradation, and close genetic similarity to the therapeutic phages indicates that these phages may be useful for therapeutic purposes for treating MDR *V. cholerae* infection in the future.

## 1. Introduction

Cholera is an acute diarrheal disease that, if left untreated, can kill a sufferer within hours due to severe uncontrolled dehydration leading to hypovolemic shock. Over time, seven pandemics of cholera have killed millions of people in every continent [1]. It has been reported that only 2 of the 200 known O serogroups of *Vibrio cholerae*, i.e., *Vibrio cholerae* O1 and O139, are pathogenic to humans due to the presence of cholera toxins [2]. Cholera pandemics are mostly produced by two biotypes of *V. cholerae* O1 strains, classical and ElTor, which exhibit comparable symptoms. Since the cholera pandemic started in 1961, the *Vibrio cholerae* O1 biotype ElTor has become increasingly specialized and is the cause of the ongoing pandemic [3]. The recent cholera outbreaks triggered by *Vibrio cholerae* O1, documented in Haiti, resulted in at least 8183 deaths and over 665,000 cases of cholera infection (https://www.cdc.gov/cholera/haiti/index.html#two, accessed on 25 January 2024). A total of 1.3–4.0 million cases of cholera arise each year, culminating in 21,000–143,000 deaths every year. (https://www.cdc.gov/cholera/haiti/index.html#two, accessed on 25 January 2024). The past decade has witnessed around 565 reported outbreaks of cholera disease resulting in 45,759 cases with 263 deaths in India, indicating the continued existence of *V. cholerae* O1 in our country [4].

The primary treatment for cholera involves Oral Rehydration Salt (ORS), and common antibiotics, notably tetracycline, fluoroquinolones, and azithromycin, have been considered to be effective in the treatment of cholera over the decades [5,6]. However, with the frequent rise of antimicrobial resistance, failures in the treatment of cholera have been reported in recent years [7]. The pathogen has acquired resistance through mechanisms such as horizontal gene transfer (HGT) involving plasmids, Integrating Conjugative Elements (ICEs), and other mobile genetic elements [8,9]. The first MDR *Vibrio cholerae* O1 strains, resistant to tetracycline, streptomycin, and chloramphenicol, were detected in the 1970s, in outbreaks like the cholera epidemic in Tanzania (1977–1978) and the Rwandan refugee crisis (1994), where 12,000 deaths highlighted the growing problem of antibiotic resistance in cholera [10].

The evolution of antibiotic resistance in *Vibrio cholerae* is further exacerbated by biofilm formation, which offers various benefits to the pathogen such as enabling survival in harsh environments, increasing antibiotic resistance, creating an appropriate system for horizontal gene transfer, and developing bacterial crosstalk pathways for efficient quorum sensing [11]. It has been found that biofilm formation also helps *Vibrio cholerae* in acquiring antibiotic resistance, which has recently been observed in *Vibrio cholerae* O1 El-Tor. Biofilm formation regulates the overexpression of pathogenic genes, limiting the effects of antimicrobial medicines [12]. *Vibrio cholerae* can build biofilm during infection, producing a hyper-infective, antimicrobial-resistant phenotype [12]. According to the report by the NIH, 80% of chronic infections and 60% of bacterial infections entail biofilm formation, which complicates the condition of the patient [13].

A failing economy and a growing country face a range of sanitation and healthcare challenges, with cholera being a serious concern that is mostly underreported [14]. Antimicrobial resistance (AMR) is considered a major future challenge, which may seriously affect 10 million people annually by 2050, leading to economic hardship of USD 100 trillion [15]. Resistance to ampicillin, nalidixic acid, chloramphenicol, and tetracycline increased over time, and the majority of clinical isolates are now resistant to almost all commonly used antibiotics, rendering treatment more difficult [16]. Despite the emergence of multi-drug resistance (MDR) and the severe pathogenic effects of cholera, it is still considered one of the world’s underreported infections [17]. The excessive and unethical use of numerous antibiotics in humans and animals has resulted in the formation of antimicrobial-resistant bacterial strains, which are now regarded as one of the most serious threats to human health [18]. This emphasizes the necessity of developing novel treatments and practicing greater antimicrobial stewardship in order to treat cholera and other diseases caused by resistant pathogens.

Over the past century, bacteriophages have been considered a natural anti-bacterial source on the earth. Phages are considered a natural form of precision medicine that has coevolved with the surrounding pathogenic bacteria. They are considered one of the most abundant creatures on the planet, and each species has evolved to battle a specific bacterial species [19]. Since, antibiotics are rapidly losing their effectiveness, many researchers believe that bacteriophages offer a promising therapy against multidrug-resistant bacteria [20]. Hence, a continuous search for novel phages with high lytic potential, particularly distinct serovars, is undoubtedly necessary for practical deployment in the treatment of superbug infections [21]. With the advancement of molecular detection, genomic characterization, and more experimental evaluation, it is possible to select a proper lytic phage for limiting a specific pathogenic bacterium [22]. Although a lot of success stories and clinical trials are going on with lytic phages against MDR pathogens, very few of them come to the forefront for therapy purposes due to a lack of well-established characterization [23]. To address the challenge posed by multi-drug resistant (MDR) infections, we are isolating different lytic phages specific to pathogenic strains of cholera. Our objective is to establish a comprehensive phage bank that will serve as a resource for effective disease management in the future, minimizing the dependence on antibiotics. In this effort, we have isolated a lot of different phages specific to *Vibrio cholerae* O1 ElTor (GenBank Id-OR420688.2, OR039881.1, OR233736.1, and OR248150.1), which can be used prior to experimental validation in future. In our study, we have isolated and characterized two phages, one from an environment sewage sample of Majhdia, Nadia region of West Bengal (Vibrio phage MJW, GenBank Id-OR248150.1), and one from the clinical stool samples of the cholera patients obtained from IDBG hospital, Kolkata (Vibrio phage 4141, GenBank Id-OR233736.1). In this study, we have performed morphological and physiochemical characterization of the isolated phages, for example, the temperature, pH, growth curve, and TEM. Bacterial killing at different MOIs, a lower latent period, and a high burst size indicate that the phages are highly virulent and lytic in nature against their host. Moreover, biofilm degradation assays by both phages strongly suggest their therapeutic nature against *Vibrio cholerae* O1 Eltor strain N16961. The genotypic basis of characterization including WGS and its bioinformatics finding suggested 4141 phage and MJW phage is under the well-characterized family of *Autographiviridae* and *Zobelliviridae*. Recent studies have revealed that *Autographiviridae* family bacteriophages are suitable for phage therapy against Klebsiella pneumoniae infection in humans [24]. *Autographiviridae* family phages are well established for anti-staphylococcal cocktail phage therapy [25]. *Zobelliviridae* family phages are rare and newly introduced family of bacteriophages that have different modes of infection strategies, and they are highly stable at different physiochemical parameters [26,27]. Studies have shown that phage cocktails can be effective in both in vitro and in vivo models against *V. cholerae*. For instance, some studies have tested combinations of lytic phages specific to *V. cholerae*, demonstrating a significant reduction in bacterial counts in experimental cholera infection models [28]. Although the 4141 phage was isolated from the clinical stool sample and the MJW phage was isolated from the environmental sewage sample, their presence in diverse systems indicates the presence of pathogenic *Vibrio cholerae*. Despite the diverse isolation system (clinical and sewage samples), both the phages showed high lytic potential in our study, which suggested the therapeutic potential against pathogenic cholera strains. Altogether, our findings indicate that both the phages have a selective host range, a broad spectrum of lytic ability, and high efficiency in biofilm degradation, which, combined with their genetic similarity to known therapeutic phages, suggest their potential role as therapeutic agents and their use for cocktail phage therapy to treat MDR *Vibrio cholerae* infections in the near future.

## 2. Methods

### 2.1. Bacteria Isolation

Fifty cholera-positive stool samples were collected from the patients of various hospitals in India (Kolkata, Delhi, Gujrat, Mallapuram, Amhedabad, Rajkot, and Karnataka) with their legal consent. Isolated samples were propagated and biochemically characterized by using thiosulfate–citrate–bile–salt (TCBS) agar plating. Positively selected colonies were grown in Luria broth (LB) and stored at −80 °C. The bacterial strains used in the recent study were grown in selected media (TCBS) using the standard methodology. Other than *V. cholerae* O1, the *V. cholerae* O139 and Non O1-Non-O139 strains were also used for phage host range determination. Moreover, selected strains of *Salmonella typhi Ty2* (ATCC-700931), *Shigella flexneri* (ATCC-29903), *E. coli* (ATCC-12435) and *Pseudomonas aeruginosa* (ATCC-10145) were used to determine the host range.

### 2.2. Bacteriophage Isolation

Clinical stool samples from the IDBG hospital in Kolkata were mixed with phosphate-buffered saline (PBS) and centrifuged at 10,000× *g* for 15 min, and the supernatant was filtered with a 0.45 µm syringe filter. Sewage samples collected from nearby regions of the IDBG hospital of Kolkata and Majhdia, Nadia, were initially filtered using Whatman filter paper (Whatman Cat-1001240) followed by centrifugation at 10,000× *g* for 15 min. This was again filtered with 0.45 µm filter in a filtration unit, and 50 mL of filtrate from each sample was mixed with the 50 mL mid-log phase bacterial (OD_600_ = 0.5) culture of *V. cholerae* O1 El-Tor strain N16961. After incubating the mixture for 20 min, it was placed in a shaker incubator (37 °C at 170 rpm) and after incubation for 4 h, 8 h, 16 h, and 24 h soups were collected and centrifuged (12,000 rpm, 15 min). The supernatant was again filtered with a 0.22 µm syringe filter, and 10 µL of each sample was used to check for the presence of phages by drop assay method with the mid-log phase culture (OD_600_ = 0.6) of *V. cholerae* [29]. The appearance of a clear zone was indicative of the presence of bacteriophages. A plaque assay was performed following the standard methodology using the double agar diffusion method with overnight incubation at 37 °C. A plaque purification method was performed for concentrating a pure phage suspension [30]. An isolated plaque was added to 10 mL of mid-log phase culture and incubated at 37 °C for 18 h, and a double agar diffusion assay was performed to analyze the titter of phages. Three or more passages of infection were performed to isolate the pure stock of lytic phages. *Vibrio* phage 4141 was isolated and purified from the clinical stool sample of the IDBG hospital, Kolkata, West Bengal, while Vibrio phage MJW was isolated from the sewage sample collected from the Majhdia region of Nadia, West Bengal.

### 2.3. Phage Purification

Mid-log phase (OD-0.6) *V. cholerae* cultures were infected with the phages at a multiplicity of infection (MOI) of 0.1. Centrifugation was performed at 7500 rpm for 30 min by adding a few drops of chloroform in the lysate followed by the filtering (Millipore syringe filter 0.22 µm) of the supernatant to discard the extra impurity; this step was repeated 2–3 times to remove the extra impurity. To concentrate the phage, a clear lysate was ultra-centrifuged at 30,000 rpm for 3 h at 4 °C using a Beckman 50.2 Ti rotor. The supernatant was discarded, and 0.5 mL of 50 mM Tris-HCl (pH 7.5), 10 mM MgSo4, and 100 mM NaCl (TM buffer) were added in each tube without disturbing the pellet and were kept at 4 °C overnight. After dissolving the phage by gentle pipetting, it was filtered by a syringe filter (0.22 µm) again to remove the extra impurities. The standard overlay method was then used to determine the phage titer [31]. A gradient of cesium chloride solutions (densities of 1.3 g/mL of CsCl in 1 mL of TM buffer, density of 1.5 g/mL of CsCl 1 mL of TM buffer, and density of 1.7 g/mL of CsCl 1 mL of TM buffer) was pipetted sequentially with a capillary pipette into the bottom of a 4.4 mL Polyallomer Ultra crimp Thin-Walled centrifuge tube. Ultra-centrifuged phage suspensions of 1.0 mL were then layered on the top of the gradients, and tubes were centrifuged at 30,000 rpm for 3 h in a TH-660 swinging bucket rotor. During this time, a roughly linear gradient of cesium chloride was formed in the tubes. After centrifuge, a bluish pure band if phage layer was formed in a 2/3rd portion of the tube. It was collected by puncturing the tube, and the dialysis (MWCO 3.5 kDa Dialysis membrane, Merck PURD-35010) was performed multiple times to remove the salts from the solution.

### 2.4. Determination of the Host Range and Efficiency of Plating (EOP)

A total of 50 clinical isolates of *V. cholerae* were confirmed by thiosulfate–citrate–bile salts–sucrose (TCBS) agar plating, and they were further cultured in Luria broth (LB) in a shaker incubator (37 °C, 190 rpm). The mid-log phase culture of standard propagating strain N16961 was used for analyzing he minimum inhibitory concentration of phage by a routine test dilution technique [32]. Phages’ lytic ability over the bacterial strains was analyzed by a drop assay technique with some modifications [33]. A mid-log phase 100 µL bacterial strain was mixed with 0.8% agar and overlayed on a Luria agar (LA) plate, and after the solidification of the soft agar, a 10 µL of diluted phage (10^−5^ dilutions) was spotted on it. The outcomes were categorized based on the clarity of the observed spot and separated into three groups: clear lytic zone (++), small lytic zone (+), and no lytic zone (-). MAK757 and N16961 were used as a positive control for the experiment.

The efficiency of plating (EOP) value was calculated according to the standard methodology with some modifications [34]. Each purified phage (10^10^ PFU/mL) was serially diluted and assayed in triplicate on the bacterial lawn with incubation at 37 °C for 17–18 h. The host strain was also tested for its PFU count in triplicate with the serially diluted phage. The plaque-forming units (PFUs) on each strain were calculated. The EOP for each strain = average PFU of the strain/average PFU of the host bacteria. The EOP was classified as high lytic efficiency = 0.5 to 1.0; moderate lytic efficiency = 0.2 to <0.5; low lytic efficiency = 0.0001 to <0.2; and no lytic efficiency = <0.001 [35].

### 2.5. Determination of Suitable Multiplicity of Infection (MOI) and One-Step Growth Curve

Mid-log phase bacterial cultures were infected with serially diluted stocks of 4141 and MJW bacteriophages, resulting in MOIs ranging from 0.0001 to 100. The inoculation period was 12 h at 37 °C with an orbital shaker at 190 rpm. Phage titers at various MOIs were determined using the double-agar-layer plate method, with the experiment conducted in triplicate [36].

The effective lytic ability of isolated phages has been determined by a one-step growth curve by calculating the burst size and latent period. An early mid-log phase (OD-0.6) host (*V. cholerae* O1 ElTor) culture was grown, and 4141 and MJW phages (10^8^ PFU/mL) were added at 0.01 and 0.03 MOI, respectively. The mixture was incubated in the dark for 20 min and then centrifuged (8000 rpm for 2 min) to collect the adsorbed host. The host pellet was washed twice and then added to pre-warmed 20 mL Luria broth (LB) and immediately placed in a shaker incubator (37 °C, 190 rpm). This time point was considered as t = 0, subsamples were collected at different time points, and the virus titer was counted by the soft agar overlay method for up to 120 min [30]. All experiments were performed in triplicate. The final burst size of 4141 and MJW phage was determined by the ratio of the final amount of virus particles discharged to the initial count of the adsorbed host.

### 2.6. Phage Stability (pH and Temperature)

The experiment was carried out to evaluate the stability of bacteriophages under various temperature and pH ranges, as described earlier [37]. In this approach, 2 mL of phages with a concentration of 10^8^ plaque-forming units per milliliter (PFU/mL) were incubated for 60 min at 4 °C, 10 °C, 20 °C, 37 °C, 40 °C, 50 °C, 60 °C, 70 °C, and 80 °C with continuous shaking at 190 rpm. In order to assess pH stability, 100 µL of purified phage particles (10^8^ PFU/mL) were cultured in 900 µL of Luria broth (LB) with pH values ranging from 2 to 14. This incubation lasted 2 h at 37 °C with constant shaking. After incubation, the phage titer (PFU/mL) was determined using the double-layer agar plate method. The survival rate provides a quantitative measure of phage resilience to pH changes in this experimental design, which allows for an assessment of phage stability in response to temperature and pH changes.

### 2.7. Biofilm Degradation Assay by Microtiter Plate Assay and Confocal Microscopy

A microtiter plate assay was performed for quantitative analysis of the biofilm degradation ability of 4141 phage and MJW phage. The bacterial suspension was cultured in Muller Hinton broth (MHB) with 1% glucose supplement and kept overnight in an orbital shaker incubator (37 °C, 190 rpm). This bacterial culture was 20-fold diluted, and 20 µL of bacteria were mixed with 180 µL of MHB supplemented with 1% glucose and inoculated in 96-well flat-bottom polystyrene microplate. After 24 h of incubation at 37 °C, mature biofilm was grown [38]. Bacteriophages were added at 10^6^ PFU/mL, 10^8^ PFU/mL, and 10^9^ PFU/mL to mature biofilm and kept for 8 h at 37 °C to evaluate the biofilm degradation ability. The plates were stained with 1% crystal violet and kept at 37 °C for 15 min and washed thrice with PBS, and the biofilm degradation ability of the phage compared with control was observed [39].

A single colony of bacteria was cultured overnight and diluted 100-fold to add it to confocal dishes. Mature biofilms formed after incubating the culture for 48 h at 37 °C. Planktonic cells were removed by washing thrice with PBS, and the phages were added at 10^9^ PFU/mL. Kanamycin sulfate of 100 ug/mL was also added with control plates for comparative analysis of biofilm degradation assay. Control and treated biofilms were fixed with 4% glutaraldehyde and stained with 20 µM/mL of SYTO-9 green fluorescent nucleic acid dye for 30 min (excitation/emission: 485/501 nm). Subsequently, 200 µL per well of film-tracer SYPRO ruby biofilm matrix stain (excitation/emission: 450/610 nm) was applied for an additional 30 min. Biofilm formation was evaluated using confocal laser scanning microscopy (Zeiss LSM710, ZEISS, Oberkochen, Germany).

### 2.8. Electron Microscopy

The carbon-coated copper grids were glow-discharged for 5 min and spotted with 10 µL of purified phage particles. After being kept at room temperature for phage particle adsorption on their surface, the grids were then negatively stained with 2% uranyl acetate and washed with 10% magnesium chloride. Grids were incubated for 30 min and examined using biological transmission electron microscope (JEOL-2100, Tokyo, Japan) at an emission of 100 kV [40].

### 2.9. Genomic DNA Extraction and Restriction Pattern Analysis

The phenol–chloroform extraction procedure was used to isolate the genomic DNA of the phage [30]. In short, about 1 mL of a high-titer phage suspension (2 × 10^10^ PFU/mL) was treated with DNase-I and RNase-A and underwent treatment with proteinase K, sodium dodecyl sulfate (SDS), ethylenediaminetetraacetic acid (EDTA, at a pH 8.0), and incubated at 55 °C for 3 h. The sample was then purified by removing debris using phenol/chloroform/isoamyl alcohol (25:24:1 *v*:*v*:*v*) and chloroform/isoamyl alcohol (24:1 *v*:*v*). After washing with pre-cooled 70% ethanol, the DNA pellet was air-dried and dissolved in Tris-EDTA buffer (10 mM Tris-HCl, 1 mM EDTA, pH 8.0), and subsequently stored at 4 °C [36]. Isolated genomic DNA was then digested with restriction endonuclease enzymes (Hind-III, Pst-I, Bam-HI, Eco-RV, Xba-I, Apa-I, Nde-I, Dpn-I) as per manufacturer protocol. Restriction digestion patterns of genomic DNA was analyzed by performing 1% agarose gel electrophoresis.

### 2.10. SDS-PAGE Analysis of Phage Proteins

A 50 µL pure phage soup (PFU = 2 × 10^10^) was boiled for 5 min, and denatured phage proteins were separated by 12% sodium-dodecyl-sulfate–polyacrylamide-gel (SDS-PAGE).

Major structural proteins were observed by standard silver staining protocol with some modifications [41]. Major structural proteins of phages were fixed with 50% methanol, and 10% sodium thiosulfate was used for sensitization. Next, silver nitrate was used for staining those protein bands and developed with sodium carbonate. A stop solution (1% -H_2_SO_4_) was used to stop the reaction.

### 2.11. Whole-Genome Sequencing

Purified genomic DNA of Vibrio phage 4141 and Vibrio phage MJW was sequenced and assembled using Illumina Nexrseq-500 platform with a 2 × 150 bp paired-end DNA library preparation. After trimming the low-quality reads and adapters, high-quality reads were assembled by the CLC workbench version 9.5.2. The assemblies were further refined using G-Finisher and the closest Phage genome was predicted by homology search. Prokka software version 1.13.4 was used for gene prediction.

### 2.12. Bioinformatic Analysis

Both the phage genomes were compared with their closest phages using ViPTree genomic comparison software version 4.0 with default parameters [42]. CG-view was performed to detect the overall Proksee CG percentile and CG skew of genomes of the phages. Putative open reading frames (ORFs) were predicted using GeneMarks-2 version 14.1 (http://exon.gatech.edu/genemark/genemarks2.cgi, accessed on 25 January 2024) [43]. The ORFs were annotated by conducting a BLASTP analysis online against the non-redundant (nr) database of the National Center for Biotechnology Information (NCBI), employing an e-value cut-off of <10^−5^. The findings were further manually verified to ensure their accuracy. The Comprehensive Antibiotic Resistance Database (CARD, https://card.mcmaster.ca/analyze/rgi, accessed on 25 January 2024) and Virulence Factor Database (VFDB, http://www.mgc.ac.cn/VFs/main.htm, accessed on 25 January 2024) were searched for the presence of any antibiotic resistance gene and virulent factors, respectively, in the predicted genomes [44,45]. Predicted transfer RNA (tRNA) genes were searched using the tRNA SCAN-SE database (http://trna.ucsc.edu/cgi-bin/tRNAscan-SE2.cgi, accessed on 25 January 2024) [46].

### 2.13. Phylogenetic Analysis

The viral proteomic tree was curated using ViPTree web server version 4.0 for constructing the phylogenetic tree. The closest phage genomes of the 4141 phage and MJW phage were manually taken according to their S_G_ score (the genomic similarity score) from the reference genome, and the final tree was constructed for an understanding of the taxonomical status of the 4141 phage and the MJW phage [42,47].

Genomic sequences in FASTA format for the 4141 phage and MJW phage were submitted to the NCBI nucleotide blast database for similarity searches. The Vibrio phage’s DNA-pol I protein was utilized to perform similarity searches in the NCBI protein–protein blast database. For the phylogenetic tree analysis, we used Molecular Evolutionary Genetics Analysis (MEGA) version 10 (X) software [48]. For protein sequence alignments and phylogenetic tree constructions, the Multiple Sequence Comparison by Long-Expectation (MUSCLE) and Unweighted Pair Group Method with Arithmetic Mean (UPGMA) tools of MEGA were performed, respectively [49]. The final phylogenetic tree was constructed from the bootstrap value of 1000 and a 75% cut-off value of the condensed tree. VIRIDIC software version 3.5.2 was used for representing the intergenomic similarity and genome length ratio of the 4141 phage and the MJW phage with its reference genomes [50].

### 2.14. Genome Sequences Accession Number

The whole-genome sequencing annotations and their feature files were submitted to the GenBank database. The complete genome information of Vibrio Phage 4141 and the partial genome of Vibrio Phage MJW is available under the accession numbers OR233736.1 and OR248150.1, respectively.

## 3. Results

### 3.1. Determination of the Morphology of Vibrio Phage 4141 and Vibrio Phage MJW

Two lytic bacteriophages were isolated from clinical stool samples and environmental water samples collected from the outbreak regions of Kolkata using *V. cholerae* O1 biotype ElTor strains N16961 as standard propagating strains. The Transmission Electron Microscopy analysis of purified phage particles confirmed that Vibrio phage 4141 contains an icosahedral head and short tail. The measurements of both phages were obtained from vertex to vertex using the Image J program version 1.5.4 (*n* = 10). Vibrio phage 4141 consists of a total length of 82.98 ± 4.3, while a head dimension measurement showed a length of 62.40 ± 1.4 nm and a width of 70.84 ± 3.0 nm, while the tail length and width were measured at 20.80 ± 0.44 nm and 17.45 ± 1.8 nm (*n* = 10) (Figure 1B). An analysis revealed that Vibrio phage MJW consists of a total length of 79.13 ± 1.3 with a head length of 62.8 ± 2.30 nm and width of 63.48 ± 2.26 nm (width) and a short tail 17.44 ± 1.55 nm in length and 10.39 ± 0.67 nm in width (*n* = 10) (Figure 1D). Based on the morphological analysis of the Vibrio phages 4141 and Vibrio phage MJW, they showed a structural similarity with the Podoviridae morphotype phages.

### 3.2. Host Range Determination and Efficiency of Plating (EOP)

Host range determination indicates that both the phages in this study, the 4141 phage and the MJW phage specifically, infect and lyse the *V. cholerae* O1 biotype ElTor strains MAK757 and N16961. Very interestingly, these phages were capable of lysing *V. cholerae* O139 strains but were found to be resistant to the other Non-O1 Non-O139 *Vibrio cholerae.* Moreover, other *vibrio* species (*Vibrio parahaemolyticus* ATCC-17802, *Vibrio mimicus* ATCC-33653), *Salmonella typhi Ty2* (ATCC-700931), *Shigella flexneri* (ATCC-29903), *E. coli* (ATCC-12435) and *Pseudomonas aeruginosa* (ATCC-10145) were also observed in spot tests, and no lytic zone was observed on the lawn of these bacteria, indicating a narrow and pathogenic cholera-strain-specific lytic activity. Both the 4141 phage and the MJW phage could infect N16961 and MAK757 with clear and round-shaped plaques (Figure 1A,C). We performed a routine test dilution assay with the serially diluted (10^2^ PFU/mL to 10^10^ PFU/mL) phages on its propagating host to calculate the minimum inhibitory concentration to achieve a clear halo zone, and we concluded that 10^5^ PFU/mL is the optimum phage concentration that can make a clear halo zone on its host (Appendix A). We tested 50 *V. cholerae* O1 biotype ElTor clinical samples with 4141 and MJW phages at 10^5^ PFU/mL (minimum inhibitory concentration). Out of 50 *V. cholerae* clinical samples, the 4141 and MJW phages effectively infected 46 and 45 samples, respectively, indicating a high range of lytic ability, which may be considered a prerequisite for using them for therapeutics (Table 1).

The lytic effectiveness of the 4141 phage and MJW phage was demonstrated by their EOP values, which were >0.5 (EOP value more than 0.5 classified as high lytic efficiency) in 46 samples and 45 samples, respectively, across various clinical samples from diverse regions of India (Table 1). The host range and EOP value strongly suggested that both the phages dominate, with a high range of lytic ability, which may be considered a prerequisite for using them for therapeutics.

### 3.3. One-Step Growth Curve and Optimal MOI

Othe one-step growth curves of both phages depict a clear latent period with a large burst size, hence suggesting the proliferative and lytic characteristics of the phages. The *Vibrio* phage 4141 and *Vibrio* phage MJW show a latent period of 15 min and 12 min with large burst sizes of 142 ± 5 and 137 ± 9 particles per infected cell, respectively (Figure 2A,D).

In our study, the highest titers of phage 4141 (2.09 × 10^11^ PFU/mL) and MJW (1.98 × 10^11^ PFU/mL) were found when the phages were propagated after infecting at MOI 0.001, and the lowest phage concentration was observed at MOI 100, suggesting the strong replicative capacity of the isolated phages (Figure 3A,B).

### 3.4. Phage Stability

Both 4141 and MJW phage have a diverse range of stability in temperature, as they can survive up to 50 °C, and the highest infectivity is shown at 37 °C (Figure 2B,E). In the case of pH exposure, they are slightly more sensitive to acidic environments as they were inactive at strong acidic conditions (pH 3 or more) and survived better at slightly alkaline conditions, as they were active up to pH 10. Their optimum activity was observed at a temperature 37 °C and a pH of 7 (Figure 2C,F).

### 3.5. Antibiofilm Activity of the Vibrio Phage 4141 and the Vibrio Phage MJW

A biofilm of *V. cholerae* N16961 was developed for a period of 24 h in a microtiter plate. The phage treatment on the biofilm showed a significant level (*p* < 0.001) of biofilm degradation in 10^6^ PFU/mL, 10^8^ PFU/mL, and 10^9^ PFU/mL wells of both the phages. The highest level of biofilm degradation was observed in 10^9^ PFU/mL wells (Figure 4II(A–D)). Fluorescence microscopy images revealed that both phages significantly reduced the biofilm produced by the host. It was found that bacteriophages showed a far better ability to destroy. Very interestingly, in our experiment, Kanamycin sulfate reduced biofilm formation by only 20%, whereas the 4141 phage reduced the biofilm by more than 90%, and the MJW phage reduced biofilm formation by more than 80% (Figure 4I(A–P),III).

### 3.6. Restriction Endonuclease Digestion Pattern of Phage DNA

Restriction endonuclease digestion pattern analysis suggested that both the 4141 and MJW phages were composed of the dsDNA genome. A total of eight restriction enzymes, Hind-III, Pst-I, Bam-HI, Eco-RI, Xba-I, Apa-I, Nde-I, and Dpn-I, were used for DNA profiling of the phages. We have reanalyzed the restriction pattern of 4141 phages with the same panel of enzymes and the restriction pattern aligned with the e-digestion of the Hind III and Nde I enzyme. The initial analysis of digests on a 1% agarose gel electrophoresis revealed that 4141 and MJW phages differ among themselves (Appendix A).

### 3.7. SDS-PAGE Analysis

The structural proteins of phages were analyzed using 12% SDS-PAGE, followed by silver nitrate straining. The major structural protein bands of 4141 phages appeared at 15 kDa, 26 kDa, and 37 kDa, while minor bands were identified approximately at 39 kDa, 42 kDa, 44 kDa, 70 kDa, 110 kDa, and 130 kDa, respectively (Appendix A lane 1). The protein profile of the *Vibrio* phage MJW revealed the presence of major structural bands at 14 kDa, 16 kDa, 20 kDa, 26 kDa, 33 kDa, 35 kDa, 37 kDa, 42 kDa, and 45 kDa and minor bands were observed at approximately 30 kDa, 42 kDa, 47 kDa, 49 kDa, 54 kDa, 56 kDa, 72 kDa, 110 kDa, 130 kDa, and 210 kDa (Appendix A lane 2).

### 3.8. Genome Sequencing and Bioinformatics Analysis

The evaluation of whole-genome sequencing revealed that *Vibrio* phage 4141 contains a double-stranded DNA genome of 38,498 bp, which comprises 46 open reading frames (ORFs). On the other hand, *Vibrio* phage MJW genome comprises a double-stranded partial genome of 49,880 bp, which consists of 64 ORFs. The Proksee CG view sequence analysis of *Vibrio* phage 4141 and *Vibrio* phage MJW showed that the genomes consist of 43% G + C and 42.5% G + C content, respectively (Figure 5 and Figure 6). Further analysis revealed that *Vibrio* phage 4141 encodes all positive CDS, while the *Vibrio* phage MJW sequence analysis unfolded 24 positive-sense CDS and 40 reverse-sense CDS. Detailed information of the mentioned phages’ coding sequence positions, directions, predicted ORFs, putative functions, and their importance in the phage life cycle has been tabulated in supplementary tables (Appendix A). Some of the important proteins of *Vibrio* phage 4141 for helping in host lysis such as in the viral genome ejection in the host are peptidoglycan hydrolase (a conserved domain of PHA00368 Internal virion protein D), which is an amidase hydrolyze link between N-acetylmuramoyl and L-amino acid residue in certain cell wall glycoprotein; holin; lysozyme proteins, which were coded by ORF 1, ORF 4, ORF 6 and ORF 36. Conserved-domain search analysis revealed that ORF5 consists of the viral receptor recognition Gp 17 tail fiber protein/collar domain protein, which is similar to the bacteriophage T7 group (thepfam03906 domain, which is a large superfamily protein of cl44509 tail fiber protein) [51]. ORF 10 codes the large terminase protein or TerL, which is a DNA maturase tat oligomerizes with ATPase and packages the DNA in the phage prohead. TerL family protein is a member of the T7 group phage. Most importantly, ORF 14 codes a DNA-directed RNA polymerase that is a key feature of the *Autographiviridae* family (T7) group of phages (Conserved domain PHA00452 T3/T7 like RNA polymerase superfamily). ORF 41 codes for the head-to-tail adapter protein typically found in the T7 group of bacteriophages. ORF 42 codes T7 capsid assembly protein. These proteins were not only replicating the genomic DNA of 4141 phages but also conferring protection from host endonucleases at the time of infection. A total of 64 ORFs (24 positive strands and 40 negative strands) were coded by the *Vibrio* phage MJW, out of which only 21 ORFs had known functions. The important proteins of MJW phages that are involved in host specificity and infection were coded by ORF 1 (partial needle head and tail spike protein), ORF 4 (host specificity and host infection), and ORF 64 (partial tail spike protein). Some other important proteins coded by the *Vibrio* phage MJW were ORF 16 (Maz-G-like pyrophosphatase), ORF 36 (Porphyrin biosynthesis protein), ORF 37 (Aerobic Cob-T subunit), and ORF 48 (Tail fiber protein). The mentioned proteins not only infected their host but also had some specialized functions as they resisted abortive host cell destruction at the time of infection by coding Maz-G pyrophosphatase [52]. No tRNA or antibiotic-resistant genes were found in the genomes of the *Vibrio* phage 4141 and *Vibrio* phage MJW.

The whole genome sequence alignments of the 4141 phage and MJW phages were compared with its closest phage genome by ViPTree web server version 4.0 genomic comparison tool, which revealed that the *Vibrio* phage 4141 (GenBank-OR233736.1) has a similar genome to Vibrio phage VP4 (GenBank-NC_007149), Vibrio phage ICP3 2009 B (GenBank-HQ641341), and Vibrio phage VP3 (GenBank-JQ780163), while the *Vibrio* phage MJW (GenBank-OR248150.1) has a genomic similarity to *Vibrio* phage ICP2_2006_A (GenBank-HQ641346), *Vibrio* phage ICP2_2011_A (GenBank-KM224878), and *Vibrio* phage ICP2_2013_A_Haiti (GenBank-NC_024791) (Figure 7A,B).

NCBI database query and VIRIDIC software genome analysis representing *Vibrio* phage 4141 were most similar to *Vibrio* phage N4 (99.7% per identity and 100% coverage) and *Vibrio* phage Rostov 1 (99.85% per identity and 95% coverage), while *Vibrio* phage MJW was most similar to *Vibrio* phage Saratov 12 (98.62% per identity and 90% coverage) and *Vibrio* phage ICP2 2011-A (95.35% per identity and 97% coverage) (Figure 8A,B).

### 3.9. Phylogenetic Tree Analysis

As proteomic tree analysis is useful to designate the phylogenetic status of newly sequenced phages, we have analyzed both the genome of the 4141 phage and the MJW phage with ViP Tree software version 4.0, which computed the genome-wide related sequence similarities curated by tBLATx [42]. The closest phage genomes are curated from the main phylogenetic tree for *Vibrio* phage 4141 and *Vibrio* phage MJW represented in Figure 9 and Figure 10. ViPTree analysis revealed that *Vibrio* phage 4141 is a T7 group phage and under the family of *Autographiviridae,* while *Vibrio* phage MJW is a Zobellviridae family phage. A detail of the related phylogenetic analysis based on the S_G_ score (related genomic similarity score) of the 4141 phage and the MJW phage is attached in Supplementary Data VipTree S2 and ViPTree S3.

In our phylogenetic analysis work, we generated a phylogeny using reference genomes curated from the NCBI database based on DNA pols in their genomes. We observed that the *Vibrio* phage 4141 showed more similarity to the *Vibrio* phage Rostov 1 (Figure 11A) and *Vibrio* phage N4, whereas the *Vibrio* phage MJW has similarities with the *Vibrio* phage ICP2 and the *Vibrio* phage Saratov 12 (Figure 11B).

## 4. Discussions

In the last few decades, over time, most antibiotics have proven ineffective against cholera infections. The uncontrolled spread of drug-resistant *Vibrio cholerae* has emerged as a significant threat to humanity and public health. Phage therapy is a potent alternative method for cholera treatment since it has more therapeutic benefits than antibiotics [53].

Numerous investigations on phage treatment have recently been carried out, and many promising laboratories have failed to produce significant results in human trials [54]. Early phage treatment failed, mostly owing to a lack of fundamental information about phage–host interactions [55]. The majority of phage experiments were undertaken without knowledge of their pharmacokinetics and pharmacodynamics. The majority of phage gene activities are still speculative, and uncovering their ORF functions provided us with a root map of phage infection patterns. Temperature sensitivity, pH sensitivity, resistance genes, virulent qualities, phage-neutralizing antibodies, enzymatic degradations, and phage loss during delivery are all major problems in human phage treatment. Concurrently, compassionate phage treatment remains the last resort hope when all viable therapeutic approaches have been exhausted [56].

Considering the current threat of AMR, the search for therapeutic phages for future treatment is the need of the hour. We have isolated and characterized cholera phages and investigated the properties of the two bacteriophages. According to modern ICTV taxonomy, *Autographiviridae* family bacteriophage *Vibrio* phage 4141 and Zobellviridae family *Vibrio* phage MJW were found to be highly effective in lysing the majority of the clinical *V. cholerae* isolates, indicating the effectiveness of the phages in the future treatment of cholera [26,57,58]. During phage characterization, we found that both the 4141 phage and the MJW phages were stable at varying temperature and pH (Figure 2) ranges. Both the phages were stable at 4 °C for long time; even after 6 months, phage titers remained almost the same as those determined by plaque count. The optimum activity of the phages was observed at 37 °C and pH 7. Interestingly, both the phages showed a large burst size of 142 and 137, having a small latent period of 15 min and 12 min, respectively. Cocktail phage therapy remains a better choice for eradicating cholera earlier, and genomic analysis suggested that the studied phages can be used for cocktail phage therapy for treating critical cholera infections in the future [28]. Also, genomic data and bioinformatics analysis revealed that they are devoid of any resistant and lysogenic genes in their genome. Phages with a small latent period, a large burst size, and a lower MOI and devoid of any resistant and lysogeny genes are the properties of therapeutic lytic phages [59]. The phages investigated in this study showed highly significant biofilm-destruction properties, indicating that they could be a better treatment option for MDR bacterial infection. Most biofilm-forming bacteria have the tendency for drug resistance, and phage susceptibility/sensitivity to those hypervirulent phenotypes is a hope for better treatment [60].

Genome sequencing analysis revealed that both the phages possess lytic genes in their genomes and have no lysogenic properties. *Vibrio* phage 4141 consisted of peptidoglycan hydrolase, holins, and lysozyme-like proteins, which reveal its robust lytic ability. The 4141 phage consists of Tail fiber, RNA polymerase, and capsid assembly proteins like the *Autographiviridae* family T7 group of phages. The *Vibrio* phage MJW consisted of a different strategy to infect their host, as it was composed of Maz-G-like pyrophosphatase, which resists host self-destruction at the time of infection [52]; Cob-T subunit, which protects the genomic DNA from host nucleases [61]; and an internal virion and tail appendage protein similar to the Zobellviridae family phages, which confers lytic ability to the phage [62].

ViPTree analysis of 4141 phage and the *Vibrio* phage MJW genome indicates that the phage is a *Autographiviridae* and Zobellviridae family phage with a similarity to Vibrio phage VP4 (GenBank-NC_007149) and *Vibrio* phage ICP2_2006_A (GenBank-HQ641346). Phylogenetic tree analysis based on DNA pol-I enzymes suggested that the 4141 phage was closely related to the *Vibrio* phage N4, as they have evolved from a common ancestor. Since *Vibrio* phage N4, Vibrio phage VP4, Vibrio phage ICP3, and *Vibrio* phage Saratov 12 are known as potential therapeutic phages, the phages investigated in this study showed almost 100% intergenomic similarity with N4 and Saratov 12, which will hopefully be better candidates as they possess other important therapeutic features like small latent period, large burst size, and wide host range and are devoid of any resistant and lysogeny genes. VIRIDIC software analysis and NCBI blast analysis for intergenomic similarity search confirmed that Vibrio phage 4141 was most similar to *Vibrio* phage N4 and *Vibrio* phage ICP 3, while *Vibrio* phage MJW phage was most similar to *Vibrio* phage ICP 2 and *Vibrio* phage Saratov 12.

In conclusion, both phages have great therapeutic potential, despite being isolated from distinct sources, i.e., environmental sewage samples and clinical stool samples. This shows that the diversity of these phages’ origins does not impede their therapeutic properties. The ongoing search for the lytic bacteriophages during seasonal diarrheal outbreaks indicates the presence of related virulent bacteria in the environment. This underscores the necessity of diverse phage-isolation approaches for effective treatment solutions. Nonetheless, additional study and validation are required to completely examine their safety and efficacy in in vitro and in vivo models prior to practical adoption.

## 5. Statistical Analysis

One-way analysis of variance (ANOVA) analysis in GraphPad Prism 8.0 was performed for data analysis. Statistical significance was *p* < 0.01 level, and the final results were plotted as mean ± standard deviation (SD).

## Figures and Tables

**Figure 1 viruses-16-01741-f001:**
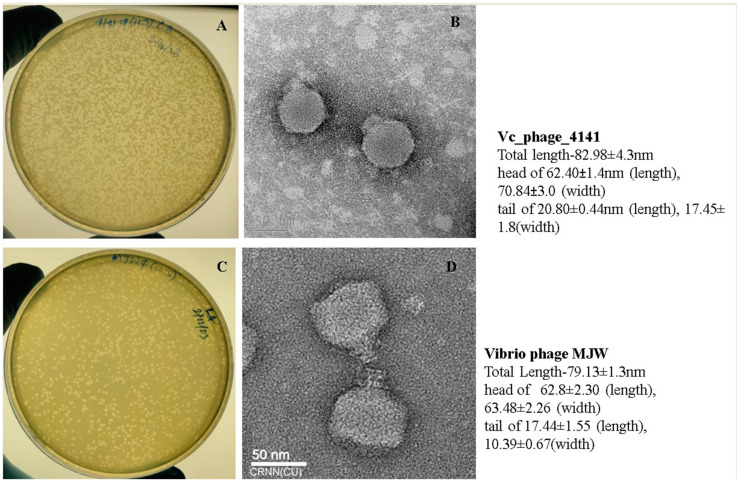
Plaques of phages on the host *V. cholerae* strain: (**A**). Vibrio phage 4141 plaque formation on the *V. cholerae* host strain; (**B**) Transmission Electron Microscopy (TEM) image of the 4141 phage particle; (**C**) Vibrio phage MJW plaque formation on the *V. cholerae* host strain (**D**). TEM image of the MJW phage. Each bar represents 50 nm in size.

**Figure 2 viruses-16-01741-f002:**
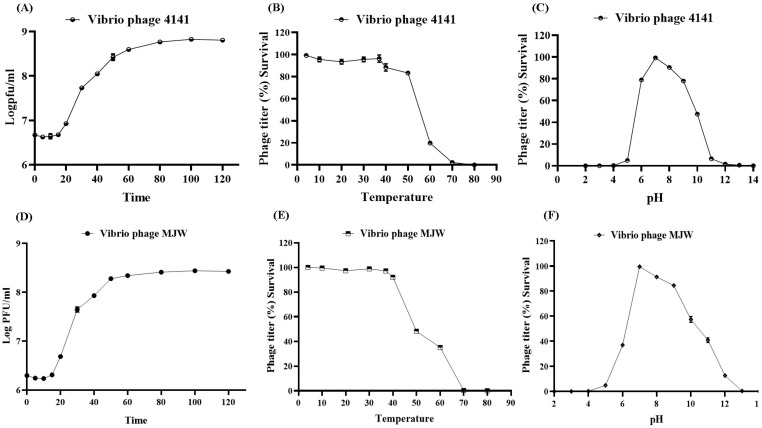
Bacteriophage growth curve and physiological stability parameters: (**A**,**D**) one-step growth experiment of Vibrio phage 4141 and Vibrio phage MJW; (**B**,**C**) and (**E**,**F**) temperature and pH stability of the 4141 phage and MJW phage, respectively.

**Figure 3 viruses-16-01741-f003:**
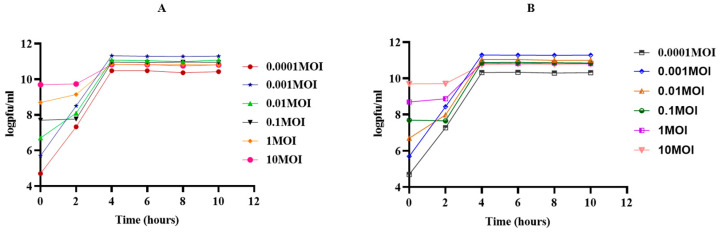
Bacterial lysis activity in vitro by phages at six different MOIs (0.0001, 0.001, 0.01, 0.1, 1, and 10): (**A**) Vibrio phage 4141 phage and (**B**) Vibrio phage MJW.

**Figure 4 viruses-16-01741-f004:**
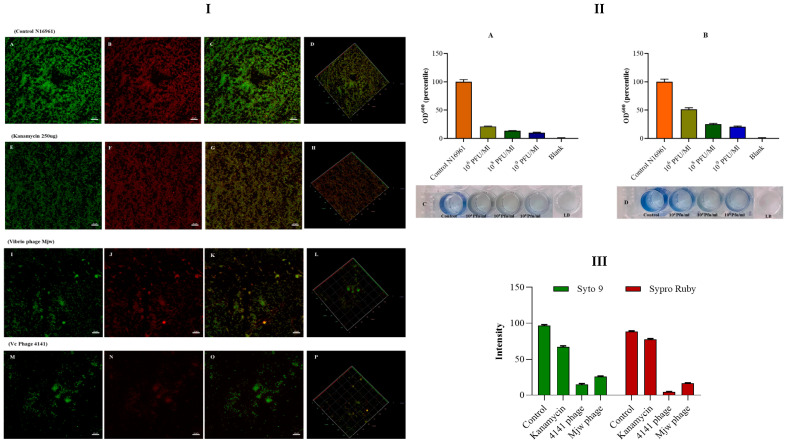
Biofilms were stained using SYTO 9 green fluorescent dye and Film-Tracer SYPRO Ruby red dye. Fluorescence images were observed using confocal microscopy. (**I**(**A**–**D**)) Control N16961 biofilm formed at 48 h; (**I**(**E**–**H**)) 200 ug/mL of Kanamycin sulfate was used to treat the biofilm; (**I**(**I**–**L**)) Vibrio phage MJW (10^9^ PFU/mL) infected and (**I**(**M**–**P**)) Vibrio phage 4141 (10^9^ PFU/mL) infected. Biofilm formation and phage treatment at three different concentrations (10^6^ PFU/mL, 10^8^ PFU/mL, and 10^9^ PFU/mL) and staining with bromophenol blue: (**II**(**A**,**C**)). Biofilm degradation by the 4141 phage (**II**(**B**,**D**)). Biofilm degradation by the MJW phage. Bar diagram represents (**III**). Fluorescence intensity of Syto 9 green and FilmTracer Ruby for control N16961, antibiotic-treated, 4141-phage-infected, and MJW-phage-infected biofilms.

**Figure 5 viruses-16-01741-f005:**
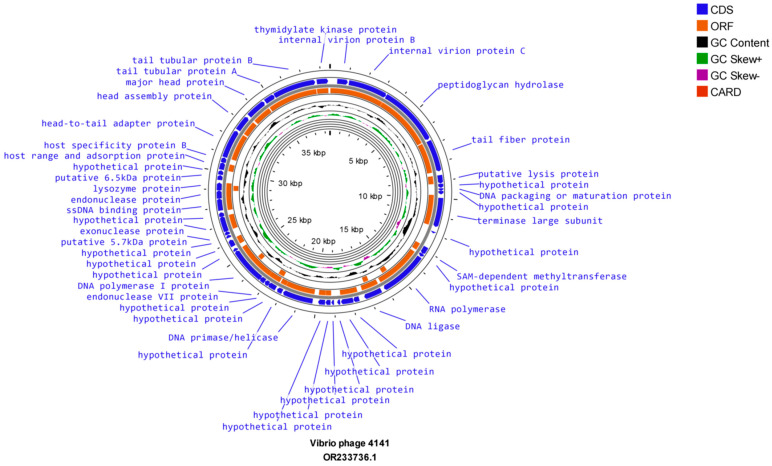
Circular map of Vibrio phage 4141: The outermost circle represents open reading frames (ORFs) encoded by the genome. The black circle represents G + C content. The outward direction indicates the G + C content of that region > average G + C content of the whole genome. Green and purple circles represent the GC-skew (G + C/G − C), and the innermost circle represents the presence of resistance genes.

**Figure 6 viruses-16-01741-f006:**
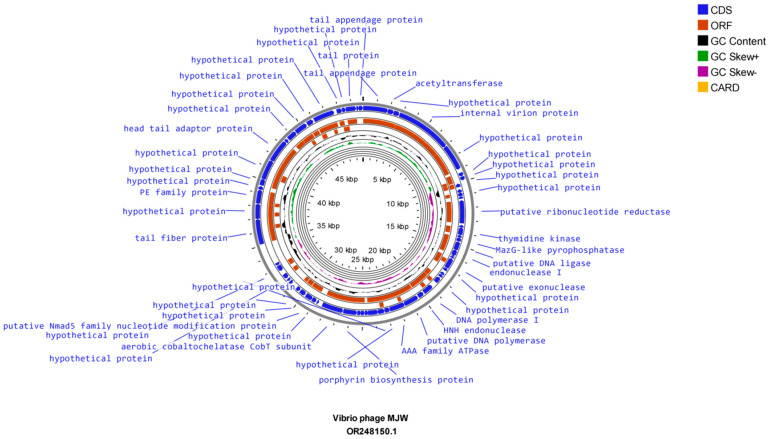
Circular map of Vibrio phage MJW: The outermost circle represents open reading frames (ORFs) encoded by the genome. The black circle represents G + C content. The outward direction indicates the G + C content of that region > average G + C content of the whole genome. Green and purple circles represent the GC-skew (G + C/G − C), and the innermost circle represents the presence of resistance genes.

**Figure 7 viruses-16-01741-f007:**
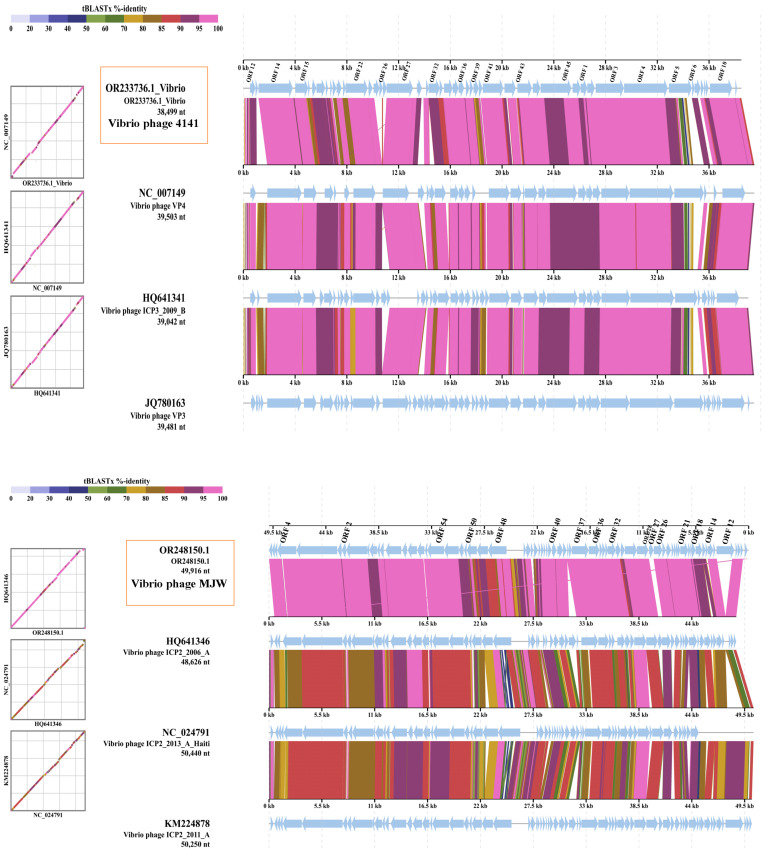
**The** ViPTree genomic comparison tool was used to compare the whole-genome sequence alignments of the Vibrio phage 4141 and the Vibrio phage MJW. (**A**) The results showed that the phage 4141 (GenBank-OR233736.1) is similar to Vibrio phage VP4 (GenBank-NC_007149), Vibrio phage ICP3 2009 B (GenBank-HQ641341), and Vibrio phage VP3 (GenBank-JQ780163), and (**B**) the Vibrio phage MJW (GenBank-OR248150.1) has genomic similarity with the Vibrio phageICP2_2013_A_Haiti (GenBank-NC_024791), the Vibrio phage ICP2_2006_A (GenBank-HQ641346), and the phage ICP2_2011_A (GenBank-KM224878). T designated colors represent the tBLASTx% (intergenomic similarity) identity of the genomes.

**Figure 8 viruses-16-01741-f008:**
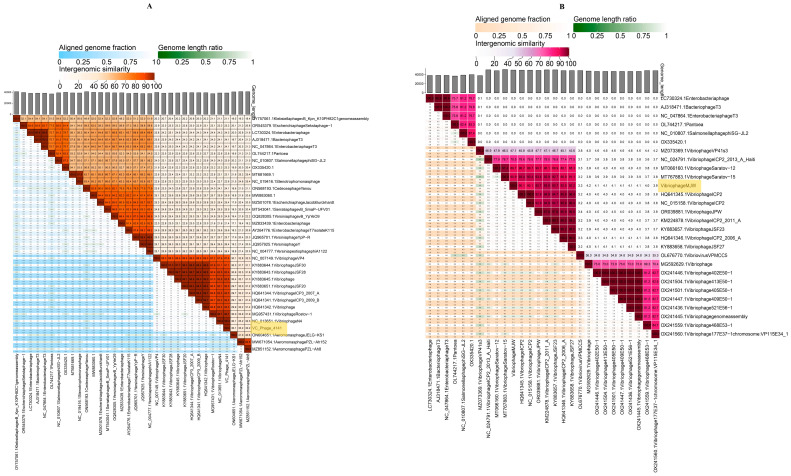
The VIRIDIC heatmap: (**A**) Vibrio phage 4141 and (**B**) Vibrio phage MJW. Reference genomes were curated from NCBI database against the mentioned phage genomes. Orange and violet shades represent the intergenomic similarity of the Vibrio phage 4141 and the Vibrio phage MJW spanning from 0% to 100%, respectively. Each cell corresponds to a specific genome comparison, with the color intensity reflecting the degree of similarity or the ratio.

**Figure 9 viruses-16-01741-f009:**
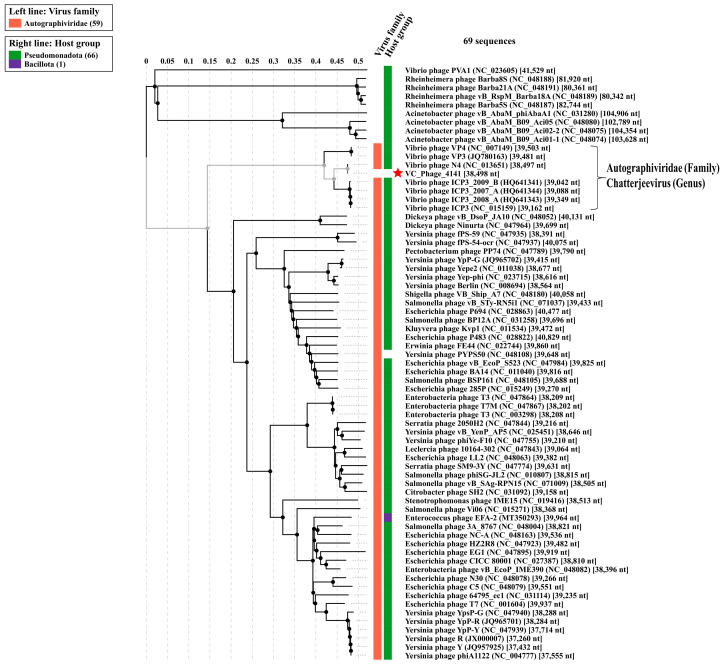
ViPTree analysis of Vibrio phage 4141: The closest phage genomes were curated according to their SG score (genomic similarity), and the final phylogenetic tree was constructed, which suggested a strong similarity with *Autographiviridae* family phages (Red * = Vibrio phage 4141).

**Figure 10 viruses-16-01741-f010:**
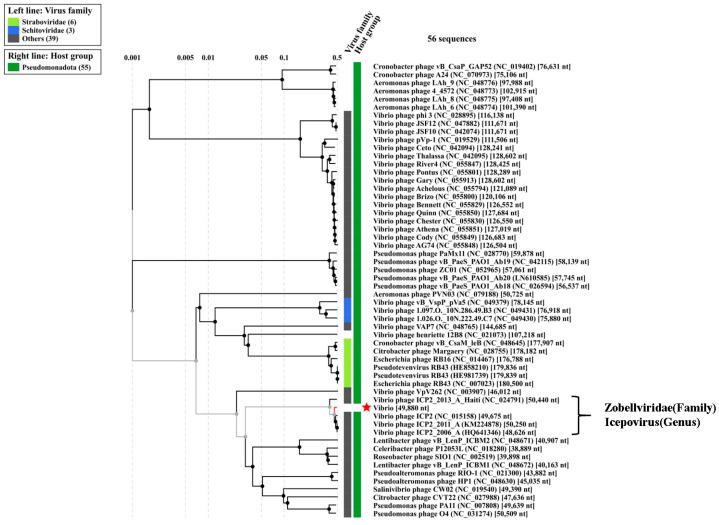
ViPTree analysis of Vibrio phage MJW: The closest phage genomes were curated according to their SG score (genomic similarity), and the final phylogenetic tree was constructed according to their similarity index, which suggested strong similarity with *Zobelliviridae* family phages (Red * = Vibrio phage MJW).

**Figure 11 viruses-16-01741-f011:**
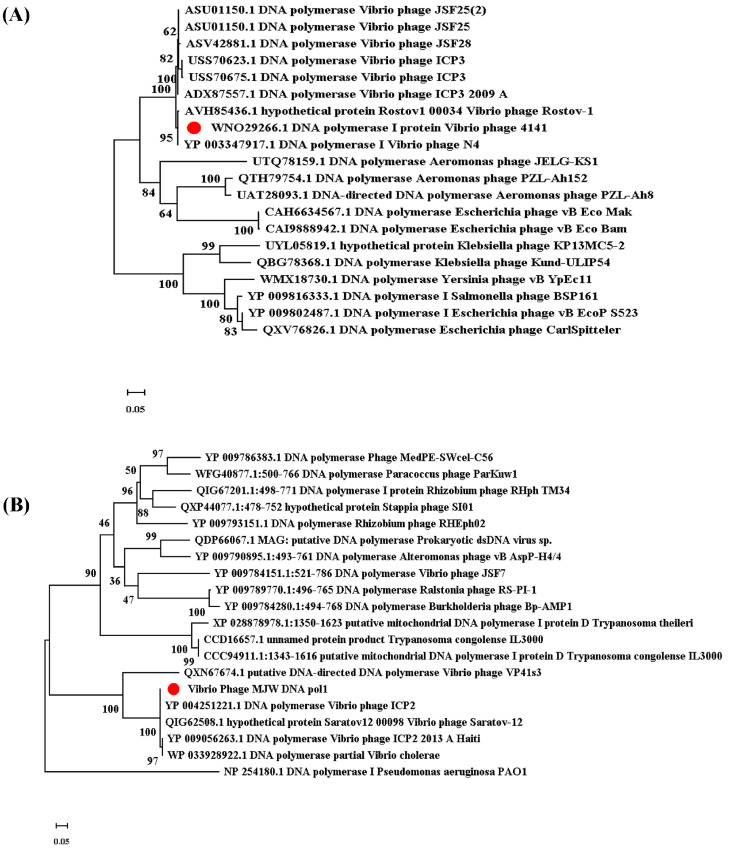
Phylogenetic tree analysis of the 4141 phage and the MJW phage. The phylogenetic tree was constructed using the maximum likelihood method with a 1000 bootstrap value. (**A**) Phylogenetic tree constructed using DNA pol I enzyme of Vibrio phage 4141 and (**B**) phylogenetic tree constructed using the DNA pol I enzyme of the Vibrio phage MJW.

**Table 1 viruses-16-01741-t001:** Host range of the isolated phages on *V. cholerae* clinical samples.

Serial Number	Sample Number	State	Year	4141 Phage	MJW Phage	EOP of 4141 Phage	EOP of MJW Phage
1	VC-314	Narendra modi hospital, gujrat	2023	++	++	1.0	0.98
2	VC-317	Narendra modi hospital, gujrat	2023	++	++	1.0	0.81
3	VC-272	Narendra modi hospital, gujrat	2023	++	++	1.0	0.68
4	VC-288	Narendra modi hospital, gujrat	2023	+	+	0.87	0.89
5	VC-321	Narendra modi hospital, gujrat	2023	-	-	0.08	0.04
6	VC-274	Narendra modi hospital, gujrat	2023	++	++	1.01	0.88
7	VC-344	Narendra modi hospital, gujrat	2023	++	++	1.05	0.91
8	VC-268	Narendra modi hospital, gujrat	2023	++	++	1.0	1.0
9	VC-319	Narendra modi hospital, gujrat	2023	++	++	1.1	0.77
10	VC-301	Narendra modi hospital, gujrat	2023	++	++	1.0	1.01
11	SPHL-14	Narendra modi hospital, gujrat	2023	++	++	1.02	0.99
12	SPHL-13	Narendra modi hospital, gujrat	2023	++	++	1.08	0.92
13	VC-01KK23	Kolar, karnataka	2023	++	++	1.03	1.0
14	IS-07	Mallapuram hospital	2023	+	-	0.51	0.44
15	IS-04	Mallapuram hospital	2023	++	++	1.01	1.02
16	IS-05	Mallapuram hospital	2023	++	++	1.04	0.92
17	IS-06	Mallapuram hospital	2023	++	++	1.0	0.81
18	IS-12	Mallapuram hospital	2023	++	++	1.0	0.78
19	IS-13	Mallapuram hospital	2023	++	++	0.98	0.93
20	IS-11	Mallapuram hospital	2023	++	+	0.94	1.0
21	IS-10	Mallapuram hospital	2023	++	++	0.91	0.69
22	IS-09	Mallapuram hospital	2023	++	++	0.89	0.83
23	IS-17	Mallapuram hospital	2023	++	++	0.79	0.97
24	VC-65/D-37	Delhi aiims	2022	++	++	1.0	1.0
25	VC-116/D-37	Delhi aiims	2022	++	++	1.02	1.06
26	VC-26/D-37	Delhi aiims	2022	++	++	1.01	0.82
27	VC-48/D-37	Delhi aiims	2022	++	++	1.0	0.87
28	VC-149/D-37	Delhi aiims	2022	++	++	0.99	1.0
29	VC-281/D-37	Delhi aiims	2022	++	++	0.91	1.0
30	VC-242/D-37	Delhi aiims	2022	+	-	0.39	0
31	VC-0095/D-37	Delhi aiims	2020	+	+	0.44	0.21
32	VC-8939	Rajkot/asalim	2021	++	++	1.0	1.0
33	VC-240	Amehdabad	2021	++	++	0.71	0.63
34	VC-26/AMH	Amehdabad	2021	++	++	0.89	0.75
35	VC-274/AMH	Amehdabad	2021	+	+	0.94	1.01
36	VC-284/AMH	Amehdabad	2021	-	++	1.04	1.0
37	VC-15/AMH	Amehdabad	2021	+	+	0.48	0.33
38	Vc-455/AMH	Amehdabad	2021	++	++	1.02	1.0
39	PL-207/GTPR	Kolkata	2019	++	++	1.0	0.88
40	PL-453/GTPR	Kolkata	2019	++	++	1.0	1.0
41	VC-0094/D-37	Delhi aiims	2020	++	++	1.1	1.02
42	VC-0084/D-37	Delhi aiims	2020	++	++	1.0	1.01
43	VC-0079/D-37	Delhi aiims	2020	++	++	1.02	1.0
44	VC-0093/D-37	Delhi aiims	2020	++	++	1.0	0.95
45	PL-161/GTPR	Kolkata	2019	-	-	0.07	0
46	PL-159/GTPR	Kolkata	2019	-	-	0	0
47	PL-211	Kolkata	2019	+	+	0.69	0.55
48	G-54/GTPR	Kolkata	2019	++	++	1.0	0.69
49	PL-471/GTPR	Kolkata	2019	++	++	1.0	0.76
50	VC-0087/D-37	Delhi aiims	2020	++	++	1.1	0.83
51	MAK757-ELTOR OGAWA	Vibrio phage laboratory		++	++	1.0	1.0
52	N16961-ELTOR INAWA	Vibrio phage laboratory		++	++	1.0	1.0

(++) = Clear lytic zone; (+) = Small lytic zone and (-) = No lytic zone formed. Growth conditions for all samples were Luria agar at 37 °C. EOP for each strain = Average PFU of the strain/average PFU of the host bacteria. EOP was classified as high lytic efficiency = 0.5 to 1.0; moderate lytic efficiency = 0.2 to <0.5, low lytic efficiency = 0.0001 to <0.2, and no lytic efficiency = <0.001.

## Data Availability

The datasets used and/or analyzed during the current study are available from the corresponding author upon reasonable request.

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
