# Peer review of "Biological Characterization and Evaluation of the Therapeutic Value of Vibrio Phages 4141 and MJW Isolated from Clinical and Sewage Water Samples of Kolkata"

_viruses, 2024, doi:10.3390/v16111741_

Round 1
Reviewer 1 Report (Previous Reviewer 4)
Comments and Suggestions for Authors
The manuscript has undergone significant revisions and, after minor deficiencies have been corrected, can be published.
Line 41 - Please change the capital letter to lowercase in the word
Lines 264, 408, 456, 457 and elsewhwere - Please italicise “Vibrio”
Please italicise “Autographiviridae” and all names of viral taxa ALL THROUGH the manuscript
“The important proteins of MJW phages which are involved in host lysis were coded by ORF 4 (Host specificity and host infection), ORF 16 (Maz-G-like pyrophosphatase), ORF 36 (Porphyrin biosynthesis protein), ORF 37 (Aerobic Cob-T subunit), ORF 48 (Tail fiber protein), ORF 62 (Host infection protein) and ORF 64 (Host infection).” - These proteins are NOT directly involved in host lysis, unlike holin, spanin and endolysin.
“AAA family ATPase and head tail adaptor protein respectively which are involved in dsDNA synthesis, packaging and overall morphogenesis of the MJW phage.” Head tail adaptor protein is a structural protein, it is not involved in DNA synthesis. Also, please check the function of AAA family ATPase - are you sure it is not the large subunit of terminase? If you want to be precise, you can just remove this sentence.
“DNA polymerase enzymes of bacteriophages are the most diversified in nature; as natural
enemies of bacteria, they evolved genome synthesis machinery with conserved and unexpected
enzymatic activity, allowing them to quickly breach the host immune response” - I would suggest deleting this sentence. It does not belong to results.
Discussion section:
“...public health.'Phage therapy'...” - Please insert space after the dot. "Phage therapy" can be written without quotation marks.
Author Response
Reviewer 1
We are grateful to the reviewer for investing valuable time in reviewing our manuscripts. Your insightful comments greatly helped us in revising the manuscript, which we believe have enhanced the quality of the manuscript. We have carefully gone through all the comments and addressed each of the comments in a sequential manner.
Comment 1:
Comments and Suggestions for Authors
The manuscript has undergone significant revisions and, after minor deficiencies have been corrected, can be published.
Response 1: Thank you very much for your time in reviewing the manuscript and suggestions for inproving the quality of the manuscript. We have corrected the minor deficiencies commented by the reviewers and responded to all of the comments. .
Comment 2: Line 41 - Please change the capital letter to lowercase in the word
Response 2: As per the comment, we have corrected the word from uppercase to lowercase in line 41.
Comment 3:
Lines 264, 408, 456, 457 and elsewhwere - Please italicise “Vibrio”
Response 3: We are thankful for your critical look in the manuscript.. We have italicize all “Vibrio” in the revised manuscript.
Comment 4:
Please italicise “Autographiviridae” and all names of viral taxa ALL THROUGH the manuscript
Response 4:
Thanks again for indicating our mistake. As per your comment, we have italicize all names of viral taxa in the revised manuscript.
Comment 5:
“The important proteins of MJW phages which are involved in host lysis were coded by ORF 4 (Host specificity and host infection), ORF 16 (Maz-G-like pyrophosphatase), ORF 36 (Porphyrin biosynthesis protein), ORF 37 (Aerobic Cob-T subunit), ORF 48 (Tail fiber protein), ORF 62 (Host infection protein) and ORF 64 (Host infection).” - These proteins are NOT directly involved in host lysis, unlike holin, spanin and endolysin.
Response 5:
Thanks for the valuable comment. As per the suggestion, we have gone through the NCBI-Conserve Domain Search and NCBI-Blast analysis of mentioned ORFs and corrected the mistake made by us in the revised manuscript. Since ORF16, ORF 36, ORF 48 and ORF 62 have no direct role to lyse the host, we have removed them from the sentence, while ORF 4 codes the host infection machinery. Homology detection and structure prediction tool (HHpred) revealed ORF 1 and ORF 64 which are partial sequences ,are similar to needle head and tail-spike protein which highlights the feature of podoviridae morphotype.
Based on the data analysis, we predicted that the MJW phage may have some different strategy to infect and lyse the host, where the tail appendage and internal virion proteins may be involved. However, prior to experimental validation, we have incorporated this statement in the text of our revised manuscript. We will work on this aspect in near future.
Comment 6:
“AAA family ATPase and head tail adaptor protein respectively which are involved in dsDNA synthesis, packaging and overall morphogenesis of the MJW phage.” Head tail adaptor protein is a structural protein, it is not involved in DNA synthesis. Also, please check the function of AAA family ATPase - are you sure it is not the large subunit of terminase? If you want to be precise, you can just remove this sentence.
Response 6: We are very much thankful for the insightful comment.
We analyzed ORF 54, which encodes an AAA family ATPase, using the NCBI Conserved Domain Search and the HHpred search tool. Analysis indicated that the AAA family ATPase is a member of the RecA superfamily (specifically, the KaiC/GvpD/RAD55 family). This family is generally responsible for catalyzing the hydrolysis of ATP, utilizing the energy derived from this process to perform mechanical work.
As we are uncertain if this protein is the large subunit of terminase, as per the suggestion, we have removed that statement from the text of the revised manuscript.
Comment 7:
“DNA polymerase enzymes of bacteriophages are the most diversified in nature; as natural enemies of bacteria, they evolved genome synthesis machinery with conserved and unexpected enzymatic activity, allowing them to quickly breach the host immune response” - I would suggest deleting this sentence. It does not belong to results.
Response 7: Thank you very much for your keen observation which has helped us in rectifying the mistakes of our manuscript.
We constructed a phylogenetic tree using the DNA polymerase I protein of the phages (Figure 10). Although we provided background information on the DNA polymerase enzyme, the statement did not align with our results. Therefore, we have removed that sentence from our revised manuscript.
Comment 8:
Discussion section:
“...public health.'Phage therapy'...” - Please insert space after the dot. "Phage therapy" can be written without quotation marks.
Response 8: Thank you for mentioning the mistake.
We have inserted the space after dot and removed the quotation mark from 'Phage therapy'.
Finally, thanks for your time and valuable comments. We have thoroughly addressed all of your feedback and are hopeful that the revised manuscript aligns with the standards for publication. Thank you once again for your valuable input.
Reviewer 2 Report (Previous Reviewer 3)
Comments and Suggestions for Authors
I have read the corrected manuscript. The authors have done a lot of work to correct the text. I have to clarify the comment to the text (Comment 4, Lines 392-395). It is known that WGS in most cases (for example, the presence of terminal repeats, redundancy during replication, etc.) represents the genomes of phages in the form of pseudo-circular structures, but this has nothing to do with the actual organization of genomes.
Response:
Regarding the designation of Vc_phage_4141 as having a circular genome and Vibrio phage MJW as having a linear genome, we agree that this requires clarification. Initially, we analyzed the sequence using NCBI BLAST, which showed that most genomes similar to Vibrio phage 4141 were circular. Additionally, the joining of CDS 1 (positions 38391-38498 and 1-411) in the feature file suggested a circular configuration.
However, we further analyzed the genome using CENSOR (a tool for detecting long terminal repeats), which identified terminal repeats in the genome. These results have been incorporated into the supplementary data (Table LTR-1). Given that we did not specifically check for genome redundancy during replication, structural confirmation, or perform restriction mapping, we have now removed the circular/linear designation from the sentence to avoid potential misrepresentation.
1. Response: the joining of CDS 1 (positions 38391-38498 and 1-411) in the feature file suggested a circular configuration.
It's not like that! The presence of the joining of CDS 1 (positions 38391-38498 and 1-411) in the feature file (at the pseudo-termini of sequences) suggested a pseudo-circular structure (result of WGS).
2. Response: we further analyzed the genome using CENSOR (a tool for detecting long terminal repeats), which identified terminal repeats in the genome.
You did not analyze the genome, but the pseudo-circular structure created by the assembler. This says nothing about the real termini of the genome and the presence of terminal repeats. My experience is that different assemblers add different additional repeats at the pseudo-termini of sequences. These repeats must be removed from the sequence.
But the authors removed the circular/linear designation from the sentence to avoid potential misrepresentation. I believe that the article can be published in Viruses.
Author Response
Reviewer 2
Thank you for the insightful comments and valuable suggestions which has helped us to improve the scientific merit of the manuscript. We have addressed all the relevant comments suggested by you. Kindly have a look.
Comment 1:
Comments and Suggestions for Authors
I have read the corrected manuscript. The authors have done a lot of work to correct the text. I have to clarify the comment to the text (Comment 4, Lines 392-395). It is known that WGS in most cases (for example, the presence of terminal repeats, redundancy during replication, etc.) represents the genomes of phages in the form of pseudo-circular structures, but this has nothing to do with the actual organization of genomes.
Response:
Regarding the designation of Vc_phage_4141 as having a circular genome and Vibrio phage MJW as having a linear genome, we agree that this requires clarification. Initially, we analyzed the sequence using NCBI BLAST, which showed that most genomes similar to Vibrio phage 4141 were circular. Additionally, the joining of CDS 1 (positions 38391-38498 and 1-411) in the feature file suggested a circular configuration.
However, we further analyzed the genome using CENSOR (a tool for detecting long terminal repeats), which identified terminal repeats in the genome. These results have been incorporated into the supplementary data (Table LTR-1). Given that we did not specifically check for genome redundancy during replication, structural confirmation, or perform restriction mapping, we have now removed the circular/linear designation from the sentence to avoid potential misrepresentation.
- Response: the joining of CDS 1 (positions 38391-38498 and 1-411) in the feature file suggested a circular configuration.
It's not like that! The presence of the joining of CDS 1 (positions 38391-38498 and 1-411) in the feature file (at the pseudo-termini of sequences) suggested a pseudo-circular structure (result of WGS).
- Response: we further analyzed the genome using CENSOR (a tool for detecting long terminal repeats), which identified terminal repeats in the genome.
You did not analyze the genome, but the pseudo-circular structure created by the assembler. This says nothing about the real termini of the genome and the presence of terminal repeats. My experience is that different assemblers add different additional repeats at the pseudo-termini of sequences. These repeats must be removed from the sequence.
But the authors removed the circular/linear designation from the sentence to avoid potential misrepresentation. I believe that the article can be published in Viruses.
Response 1: Thank you very much for your valuable Comments to improve the quality of the manuscript
We have analyzed the genome with Tandem Repeat Finder and also curated NCBI Blast and LTR-CENSOR web server earlier. You pointed out correctly about the assembler and they need to remove from the sequence.
As above analysis doesn’t confirm the circularity of the genome, we have already removed the circular/linear designation from the statement in our revised manuscript.
Once again, we are very much thankful for helping us with your valuable comments and suggestions to avoid potential mistakes and improving our revised manuscript to publish in the renowned journal.
Reviewer 3 Report (Previous Reviewer 2)
Comments and Suggestions for Authors
The authors have improved the manuscript in accordance with my suggestions. The only issue I still have is the clarity of host range figures, but the manuscript is fine overall
Author Response
Reviewer 3
We are very much thankful to the constructive and helpful comments, which have significantly improved the quality and clarity of our manuscript.
Comment 1:
Comments and Suggestions for Authors
The authors have improved the manuscript in accordance with my suggestions. The only issue I still have is the clarity of host range figures, but the manuscript is fine overall
Response 1: Thank you once again for your valuable time and insightful comments.
As per the suggestion provided, we have updated the figures with bigger fonts and bigger size. However, we have added the raw data and large figures in our supplementary data file. The figures have been revised and are now well-aligned with the configuration needed for publishing in scientific journal. Hope this will strengthen our submission significantly. Also, a high resolution version of the figure 2 and figure 3 have been incorporated to the supplementary file for better visualization. Kindly look into it.
Reviewer 4 Report (Previous Reviewer 1)
Comments and Suggestions for Authors
See attachment

Author Response
Reviewer 4
We appreciate for dedicating valuable time in reviewing our manuscript. We have addressed each comment carefully and sequentially. Your insights have significantly assisted us in revising the manuscript, ultimately improving the overall quality of our work.
Comment 1:
In the resubmitted manuscript some issues have been repaired:
The EM provided now looks like the same kind of phage as the sequence.
There is now a ViPtree and the taxonomic description has been corrected.
The one step growth curves now look like the results attributed to them.
Errant descriptions of genes that span the ends of the genome have been removed.
They are no longer talking about turbid plaques for lytic phages.
Response 1: Thank you for your thoughtful feedback and suggestions which improves our manuscript. Limited resources of our laboratory make it delay to address all the potential issue timely, however we have done the electron microscopic images and all the experiments within that time frame, we are very much thankful to the reviewer for dedicating the valuable time to review the manuscript.
We have addressed all the relevant issues commented in our manuscript sequentially.
Comment 2:
Some other issues are still problematic.
The introduction
I had complained that the introduction was a standard explanation that cholera was bad and the claim that phages were going to replace antibiotics any time now without any specific rationale for characterizing these specific phages. Now it reads like a much longer explanation that cholera is bad and phages are going to replace antibiotics any time now. In your response to my first review you have the sentence: "As a part of our research activity, with an intention to build phage bank for future crisis in disease management by antibiotics, we are isolating lot of different phages." That's a much better introduction to this paper than what you've got here. The introduction you have says that you isolated two phages and that they are supposed to be especially good candidates for phage therapy. If you have some references to some other phages you published in this effort or some summary number of how many you've put in the bank, that would be appropriate.
Response 2: Thank you for the valuable comment. Our laboratory is focused on isolating various lytic phages to evaluate their therapeutic potential. With the aim of establishing a phage bank in the future, we are isolating phages specific to cholera strains from various regions of India and we are experimentally validating their lytic potential.
As per your suggestion, we have introduced the phages we have isolated and submitted to GenBank in our revised manuscript. Additionally, we have revised our goals to better address the future antimicrobial crisis with the intention to establish a characterized phage bank in the revised manuscript.
Comment 3: You end the introduction by claiming that these two phages have particularly high lytic activity. But there were no controls. You didn't measure them in comparison to anything. If you do the study where you first measure lytic activity of a larger number of isolates and then pick the high ones for sequencing, then you could claim that. I didn't gather that you did that here. If you just picked a couple of isolates at random, then the lytic activity found is pretty much by definition going to be average. Everyone in this field does this thing of making comparative statements without making any actual comparisons. But if your argument is that you need more phages, not necessarily the best phages, then this exaggeration is completely unnecessary. In that regard, a reference to some review about making cocktails to support the idea that you might need a lot of phages might be appropriate.
Response 3: Thank you very much for your thoughtful comments which are extremely supporting in enhancing the scientific soundness of the manuscript.
We have evaluated both the phages lytic potential in wet lab prospective and also genomic prospect which suggested their potential therapeutic effects against cholera. Your comment matches with the need of our primary goal, we want to make a phage bank and we need more lytic phages to make a good cocktail therapy. To better reflect this goal, we have accepted your suggestion and revised the language to avoid the appearance of an unqualified comparative claim.
Also, your suggestion to reference (Citation number 29) a review on the benefits of phage cocktails to support the need for a diverse set of isolates is excellent and strengthens the overall rationale. We have incorporated your suggestive input in the introduction and discussion section (as highlighted) in our revised manuscript..
Comment 4:
The comparative figure:
The comparative figure still has their phage sequences in the wrong orientation and in one case circularly permuted relative to the references. That's what causes these criss crossing patterns. Doing this has two ill effects. 1) If there are some organizationally varied regions in this genome, you can't see them with the figure all twisted up like this. 2) It advertises that you don't know how to use the computer program. You may argue that you see a lot of this in phage papers. Sadly, that's exactly right.
I won't insist you have to fix it, but if you don't know how: First you determine if your sequence is in the same orientation to the one being compared. If not, find a program or a web site to make the reverse complement. Then make your sequence have the same gene on the left end as the genome being compared. That typically involves cutting away sequence at one end and pasting it on the other end. Then send that to whatever program or web server you're using. Those programs do not do these steps for you. If that changes the gene order from what you discuss in the paper or your GenBank file, hand edit enough gene number onto the figure so that the reader can follow the rearrangement. It's good to add some numbers even if you don't rearrange it. If you're comparing to more than one sequence, you may have to also complement and/or circularly permute others of those and send those along with your new phage.
If you do that in comparison to the phages you are using here, I suspect all you'll get back is a solid box, because I think you're comparing to very similar genomes. If that's what you want to show: that the new phages are almost identical to known phages, then fine. If not, then an alternative is to compare to something a little further away and that could be considered more of a prototype or type phage for the group. Then if you start getting some regions of recombinational variation, that might help highlight the antireceptor.
Response 4: We are very much thankful to the reviewer for deep analysis and thoughtful comments to the manuscript which makes the manuscript a better version.
We have analyzed both the genome to their closer reference genome with Clustal Omega/ Multiple sequence alignment and found the orientation of the genome. As per the reviewer suggestion we have rearranged the ORFs to compare with their closer relatives. In case of 4141 phage the closest phage got almost a solid box, for this reason we considered a less close genome to compare their relative ORFs. We have also rearranged MJW phage ORFs for better comparison and incorporated the important numbers as per respective GenBank features. We have incorporated the changes made in figure 10 (Figure 11 of the earlier version) in the revised manuscript.
Comment 5:
Restriction digestion:
In your response to my previous comments you say you redid the digestions and now "We have found that the Vibrio phage 4141 restriction profile matches with the e-digestion pattern of Hind III and Nde I restriction enzyme." That would be good to say in the manuscript. Do you now access that all the other digests shown are incomplete? If so, that would be good to mention also. Personally, I don't put failed restriction digests in papers, but if you really want to, I won't force you to take them out.
Response 5: Thank you for your valuable suggestions and indicating the issue about the restriction profile.
We have reanalyzed the restriction profile. In the earlier round of review, one of the reviewer suggested us to include the figure in the manuscript for a clearer review of the digestion patterns. As per your advice we have incorporated the suggested statement “We have found that the Vibrio phage 4141 restriction profile matches with the e-digestion pattern of Hind III and Nde I restriction enzyme” in the revised manuscript. However, we agree with you about putting the figure in manuscript. Although not removing the figure-5, we are placing the figure from the main text figure to the supplementary file as Figure-S2. for better clarity. Hope this will be fine to address your comment.
Comment 6:
Unreadable figures 10 and 12.
Figures 10 and 12 are in too low of resolution. They can't be read, even if you zoom in on them, at least in the versions sent to me. Mega allows you to change the font size. I don't use Viridic, but the manual says how to change font sizes. If the problem with the heat maps is that for the number of sequences you want, the figure is too big and you're losing resolution reducing it, then put the high resolution version in the supplement. If no other options exist, reduce the number of sequences compared. There is no point in showing a figure that can't be read.
Response 6: We are thankful for your careful observation.
We agree with you and have re-configured figure 10 and figure 12 with better resolution. We have incorporated the modified figures as figure 9 and figure 10 in our revised manuscript. Also, we put the same version of high resolution images in supplementary file for your review.
Comment 7:
Minor issues:
You have "Table 3: Host range of the isolated phages to V. cholera clinical samples." inserted as a page header repeating on every page in the copy I got.
Legend to table 3 talks about ++ and +, but in the table they are ** and *.
There seems to be a table 1 and table 3, but no table 2.
Response 7: Thank you for correcting us. The header was placed in the manuscript by mistake, we have corrected the revised manuscript and removed the header.
Actually the legend of Table 1 which looks like “ * ” marks is actually a “ + ” sign. Due to smaller fonts it looks like “ * ” marks. We have corrected the font size and made it bold for a clear visualization.
There have no Table 3. We have incorporated Table 1 in main table while Table S1 and Table S2 incorporated as supplementary tables.
Once again we are very much thankful t to you for your effort, time and support in improving the quality of the manuscript.
Round 2
Reviewer 4 Report (Previous Reviewer 1)
Comments and Suggestions for Authors
I have no further comments.
This manuscript is a resubmission of an earlier submission. The following is a list of the peer review reports and author responses from that submission.
Round 1
Reviewer 1 Report
Comments and Suggestions for Authors
This manuscript describes two new Vibrio cholera phages and are characterized as to growth properties, EM morphology, sequence, and host range. There is a lot of confusion in this manuscript.
Major issues
Vibrio phage 4141 is described in the abstract as a member of Myoviridae in order Caudovirales. ICTV no longer recognizes Myoviridae or Caudovirales. Normally I’d say that was a minor problem: just say it is morphologically a myovirus and find the current classification either by running it through the ViPtree web server, or just note the classification of a closely related phage if it exists. In your GenBank file you did acquire the classification, presumably from Vibrio phage N4. But that is Autographiviridae; Studiervirinae; Chatterjeevirus. which is a podovirus, not a myoviruses, yet the EM image shown for 4141 is clearly that of a myovirus. The worst case scenario would be that the phage wasn’t adequately purified and there are two dominant phages still in the mix. Then multiple phage contigs appear in the sequence data, multiple phages appear in the EM images, and you end up picking a different sequence contig than the particle shown in the EM figure. There’s no way to tell which phage belongs to the other characterizations reported in the paper, and after the phage is propagated, it’s unpredictable what phage will end up dominating the culture.
In principle the restriction digests shown in supplement fig. S2 should clarify this problem. The point of a restriction digest is to size the fragments and correlate them with the fragments predicted from the sequence. Then you’d also notice one of the predicted fragments is cleaved in two; that tells you were the ends of the genome are. In the case of 4141, the largest HindIII fragment is predicted to be 6 kb. There are 8-10 fragments in fig S2 bigger than 6 kb according to the size markers. So either the DNA shown belongs to a much bigger phage, or this is a partial digest. Many of the digests look to be partial. The point of the analysis is to incubate until you get a limit digest. The HindIII digest doesn’t look that much like a partial digest to me, but I don’t know.
A similar concern happens in the host range data. On some clinical isolates the result is reported to make clear plaques and on some it is reported to make turbid plaques. The sequence shown is of an obligate lytic phage. One way to create turbid plaques with a lytic phage is if the clinical isolates have not been adequately purified. Then you can be plating on a mixed culture of sensitive and resistant cells, with the resistant cells making the plaques look turbid. Another way is to have a mixture of phages with a major component being lytic and a minor component being temperate. Then on a host that the lytic phage can not infect, you start to notice a lower titer of the temperate phage making turbid plaques.
Another major problem is with the growth curves. Line 333: “Vibrio phage MJW shows a latent period of 15 minutes and 12 minutes with a large burst size of 142 and 137 particles per infected cell respectively (Figure 2A and 2C).” This is a reasonable result, but the curves shown (actually 2A and 2D) appear to indicate a burst of around 1000, which is unreasonable. I have no idea what happened there.
Other issues:
The introduction is a stock introduction mainly documenting that cholera is bad, with the standard paragraph about how phages are going to replace antibiotics. But there’s no real information about why these particular two phages would be helpful, especially considering that they are both almost identical to previously known phages. There was no literature work on the phage groups which leads to discussion of the 4141 as if no one had ever seen a T7-like phage before. If you’re going to go through the gene content gene by gene, you should get it right. Autographiviridae phages definitively have an RNA polymerase, and orf4 is an injection protein (IVPD), not a lysin. The peptidoglycan hydrolase domain on that protein is a tail lysozyme, not a protein involved in cell lysis. But honestly, you should probably just say it has the usual core gene content for Studiervirinae and just focus on features relevant to the point of the paper. That would mainly be to find and describe the antireceptor. It’s the C-terminal portion of orf5. If you’re going to make a cocktail, that’s the gene that needs to differ among the phages. Don’t just hide it in the supplement and call it “morphogenesis protein”. The gene interrupted by the end of the sequence is IVPA. It is errantly said to start on an unconventional start codon in the text, but it is correctly annotated to start on an AUG codon in the GenBank file.
MJW is a member of Zobellviridae. It’s harder to find a well annotated member of that family, but they do exist. In this case, the annotation got copied from a weakly annotated family member, and is barely recognizable. In MJW, the definitive podoviral structural protein (tubular B tail protein) is broken over the end of the sequence and is errantly said to start on an unconventional start codon in the text. The taxonomy and the position of the tubular tail B gene are correct in the GenBank file, but wrong in the paper. The fact that tubular tail B is intact across the sequence ends means that the “partial” designation given in the GenBank entry for this genome is incorrect.
The criss-crossing patterns in fig 8 are due to circular permutation and orientation errors.
I encourage all authors of new phages to submit the phages to a phage repository.
Author Response
Point to point response to the reviewer comment
Reviewer 1
Thanks a lot for investing your precious time for reviewing our manuscripts in depth. We have gone through all of your comments and addressed them sequentially. We are really thankful for your insightful comments which indeed helped us to revise the manuscript thereby improving the quality of our work.
Major issues:
Comment 1:
Vibrio phage 4141 is described in the abstract as a member of Myoviridae in order Caudovirales. ICTV no longer recognizes Myoviridae or Caudovirales. Normally I’d say that was a minor problem: just say it is morphologically a myovirus and find the current classification either by running it through the ViPtree web server, or just note the classification of a closely related phage if it exists. In your GenBank file you did acquire the classification, presumably from Vibrio phage N4. But that is Autographiviridae; Studiervirinae; Chatterjeevirus. which is a podovirus, not a myoviruses, yet the EM image shown for 4141 is clearly that of a myovirus. The worst case scenario would be that the phage wasn’t adequately purified and there are two dominant phages still in the mix. Then multiple phage contigs appear in the sequence data, multiple phages appear in the EM images, and you end up picking a different sequence contig than the particle shown in the EM figure. There’s no way to tell which phage belongs to the other characterizations reported in the paper, and after the phage is propagated, it’s unpredictable what phage will end up dominating the culture.
Response 1:
Thank you for your insightful comments on our manuscript. This will definitely help us to improve the quality of our work.
As suggested by you we have performed the ViP Tree analysis and NCBI taxonomy analysis to determine the family of the phages according to the new classification of ICTV. We have found that according to new ICTV classification Vibrio phage 4141 and Vibrio phage MJW falls under the family Autographiviridae and Zobellviridae respectively. This has been updated in the revised manuscript accordingly.
We appreciate the reviewer for his observation on the conflicting data between EM and WGS of the 4141 phage, therefore, we have repeated the experiment with of 4141 phage to ensure the accuracy. We perform a new Transmission Electron Microscopy (TEM) imaging for ensuring the purity and morphological structure of the phage. The TEM image confirmed that the morphology of the 4141 phage resembles as the member of the family Autographiviridae. The previous image of 4141 phage was wrongly incorporated in the manuscript which was probably due to the mistake in numbering of the grid. We really apologize for the inconvenience caused due the incorporation of the wrong figure. In the revised manuscript we have replaced the figure 1B with the new figure obtained by the TEM imaging accurately reflecting the phage morphology.
Comment 2:
In principle the restriction digests shown in supplement fig. S2 should clarify this problem. The point of a restriction digest is to size the fragments and correlate them with the fragments predicted from the sequence. Then you’d also notice one of the predicted fragments is cleaved in two; that tells you were the ends of the genome are. In the case of 4141, the largest HindIII fragment is predicted to be 6 kb. There are 8-10 fragments in fig S2 bigger than 6 kb according to the size markers. So either the DNA shown belongs to a much bigger phage, or this is a partial digest. Many of the digests look to be partial. The point of the analysis is to incubate until you get a limit digest. The HindIII digest doesn’t look that much like a partial digest to me, but I don’t know.
Response 2:
Thank you very much for observing every detail in our manuscript so carefully. We appreciate your comments on the restriction enzyme digestion analysis of the phages.
In order to resolve the above problem highlighted by the reviewer, we have purified the phages with the help of CsCl gradient and extracted the genome with the help of Phage genome isolation kit (Norgen Biotek). We have repeated the restriction pattern analysis of the 4141 phage genome with the same restriction enzymes sets used in our experiment previously, and, also verified the restriction pattern with in silico digestion of the Vibrio phage 4141 genome created with the help of neb cutter 2.0. We have found that the Vibrio phage 4141 restriction profile matches with the e-digestion pattern of Hind III and Nde I restriction enzyme. The new figure is inserted in the revised manuscript as Figure 5A.
Comment 3:
A similar concern happens in the host range data. On some clinical isolates the result is reported to make clear plaques and on some it is reported to make turbid plaques. The sequence shown is of an obligate lytic phage. One way to create turbid plaques with a lytic phage is if the clinical isolates have not been adequately purified. Then you can be plating on a mixed culture of sensitive and resistant cells, with the resistant cells making the plaques look turbid. Another way is to have a mixture of phages with a major component being lytic and a minor component being temperate. Then on a host that the lytic phage can not infect, you start to notice a lower titer of the temperate phage making turbid plaques.
Response 3:
We are thankful for your valuable comments on our manuscript.
For the determination of the host range, we have performed spot assay with the minimum inhibitory concentration (10-5) dilution from the stock phage that was determined with the reference strain N16961 was used. Previously, we have investigated the MIC with routine test dilution technique and observed that 10ul drop from 10-5 dilution i.e 105 PFU/ml phage was enough to make clear lytic zone when spotted on its host N16961. There were no turbid plaques in any clinical strain, 46 strains were sensitive to 4141 phage while 45 samples were sensitive to MJW phage. In case of 4141 phage few strains which we designated as ‘+’ resulted in smaller lysis zone and we have written as opaque lysis or turbid plaques, which is a mistake from our side, actually it should be designated as smaller lytic zone. We have rectified our error and incorporated the corrections in the manuscript and updated table 1.
For further validation of our data, we have performed the Efficiency of Plating (EOP) of both the phages with clinical strains of V. cholerae and observed that 46 clinical strains in case of 4141 phage have shown a minimum EOP >0.5. Whereas 45 samples in case of MJW phage also showed a minimum EOP >0.5. We didn’t find any turbid plaques while performing the EOP that clearly suggest the purity and highly lytic efficiency of the phage.
We are grateful to the reviewer for suggesting us to perform the EOP of both the phages. This suggestion has help to improve our manuscript.
Comment 4:
Another major problem is with the growth curves. Line 333: “Vibrio phage MJW shows a latent period of 15 minutes and 12 minutes with a large burst size of 142 and 137 particles per infected cell respectively (Figure 2A and 2C).” This is a reasonable result, but the curves shown (actually 2A and 2D) appear to indicate a burst of around 1000, which is unreasonable. I have no idea what happened there.
Response 4:
Thank you very much for your keen observation which will help us to rectify our manuscript thereby enhancing the quality of our work.
As highlighted by the reviewer, we have gone through the Figure 2A and Figure 2D, and, found that the reviewer had correctly identified the errors in the figures mentioned above. In this regard, we re-evaluated the data and understood that we made a mistake while analysing the results. We have re-analysed the data and incorporated the corrected images in the new manuscript as Figure 2A and 2D.
Thank you once again for helping us to improve our manuscript, we wholeheartedly recognize your contribution.
Other issues:
Comment 5:
The introduction is a stock introduction mainly documenting that cholera is bad, with the standard paragraph about how phages are going to replace antibiotics. But there’s no real information about why these particular two phages would be helpful, especially considering that they are both almost identical to previously known phages. There was no literature work on the phage groups which leads to discussion of the 4141 as if no one had ever seen a T7-like phage before. If you’re going to go through the gene content gene by gene, you should get it right. Autographiviridae phages definitively have an RNA polymerase, and orf4 is an injection protein (IVPD), not a lysin. The peptidoglycan hydrolase domain on that protein is a tail lysozyme, not a protein involved in cell lysis. But honestly, you should probably just say it has the usual core gene content for Studiervirinae and just focus on features relevant to the point of the paper. That would mainly be to find and describe the antireceptor. It’s the C-terminal portion of orf5. If you’re going to make a cocktail, that’s the gene that needs to differ among the phages. Don’t just hide it in the supplement and call it “morphogenesis protein”. The gene interrupted by the end of the sequence is IVPA. It is errantly said to start on an unconventional start codon in the text, but it is correctly annotated to start on an AUG codon in the GenBank file.
Response 5:
We appreciate the insightful feedback provided by the reviewers. We have carefully revised our manuscript to address all the highlighted concerns. The response are as follows:
As suggested by the reviewers we have thoroughly revised the introduction, moving beyond the generic focus on phage therapy and providing a more detailed justification for the relevance of the two phages under study. The updated section now highlights their potential applications and discusses their unique features in the context of existing phage research.
We have re-annotated the genomes as recommended by the reviewer. Specifically, we identified orf4 as the IVPD injection protein, which contains a tail lysozyme domain, and orf5 as a tail fiber and collar domain protein that includes a receptor recognition component. This revised annotation is reflected in Table S1 of the manuscript. Additionally, we have used the NCBI Conserved Domain search to analyze the conserved domains, focusing on the most relevant features for phage characterization. Further, we have also updated the feature files to standardize start codons.
Following your suggestion, we performed a classification of both phages using the ViPTree web server. We have also conducted a comparative analysis of the phage genomes against their closest relatives. These results are presented in the revised manuscript as Figure 8, Figure 9 (ViPTree analysis of 4141 and MJW phages), and Figure 11 (genomic comparison with the closest phages).
We believe that these changes significantly strengthened the manuscript and hope that it now meets the expectations of the reviewers. Thank you once again for your valuable comments and suggestions.
Comment 6:
MJW is a member of Zobellviridae. It’s harder to find a well annotated member of that family, but they do exist. In this case, the annotation got copied from a weakly annotated family member, and is barely recognizable. In MJW, the definitive podoviral structural protein (tubular B tail protein) is broken over the end of the sequence and is errantly said to start on an unconventional start codon in the text. The taxonomy and the position of the tubular tail B gene are correct in the GenBank file, but wrong in the paper. The fact that tubular tail B is intact across the sequence ends means that the “partial” designation given in the GenBank entry for this genome is incorrect.
Response 6:
We are thankful to the reviewer for introducing the issue.
As highlighted by the reviewer about the problem in gene annotation, we have re-annotated the genome and updated the manuscript and the figures accordingly. The phage got partial genome designation as it codes a partial gene at each end of the genome resembling that the tubular B tail protein is broken over the end of the sequence which we have indicated in our text. Taxonomy and the position of the ORFs were updated as per GenBank file of both the phages and respective details have updated in Table S1 and Table S2.
Comment 7:
The criss-crossing patterns in fig 8 are due to circular permutation and orientation errors.
Response 7:
Thank you for your valuable and deep analysis of our manuscript to improve it.
As highlighted by you, we have reanalysed the data with ViPTree genomic comparison tool and updated the figure 8 which is now figure 11 in our revised manuscript.
Comment 8:
I encourage all authors of new phages to submit the phages to a phage repository.
Response 8:
Thank you for encouraging us for submitting phages to phage repository.
We will definitely submit our well characterised phages to the phage repository. As a part of our research activity, with an intention to build phage bank for future crisis in disease management by antibiotics, we are isolating lot of different phages. We are in touch with ICTV, India in this regard. Phage society needs to cooperate each other to improve the domain of phage biology. Thank you very much.
This is a real need, let us hope very positive save several lives in future
Finally, thanks for your time and valuable comments. We have responded to all of your comments and hope the revised manuscript is up to the standard of publication.
Reviewer 2 Report
Comments and Suggestions for Authors
Line 58 O1 is in italics
Introduction - the part about V. cholerae and it's pathogenicity and antibiotic resistance is a bit "ragged". Some sentences have no connection with one another or any logical train of thought. I.e. Lines 62-67 Do the authors mean the resistance of V. cholerae or resitance in general as the numbers suggest it refers to general problem. So how does it apply to V. cholerae? Then lines 68-74 seem to answer the question, but it also make the previous sentence redundant.
Methods 2.1 Bacteria isolation - please name the hospitals or at least the cities in which te hospitals were localised as from this description it is impossible to know the scale of the isolation (global? local?). The names are written in the results section but it should also be pointed out in the methods. Also, what strains of Enterobacteriales were used for host range? Any ATCC reference strains or some clinical isolates? There is no mention of those strains characteristics anywhere in the manuscript
2.2 Bacteriophage isolation: again, name the cities or sewage treatment plants from which the samples were obtained, especially since you mantion that the samples from Kolkata were treated differently than the rest of the samples.
Line 137 V. cholera cultures was - V. choleraE cultures WERE
2.4 Host range - the separation using the plaque morphology or just the titer is no logner considered informative. Instead the EOP should be used
2.6. Phage stability (pH and Temperature) a mistake in citation number - 454?!
2.9. Genomic DNA extraction and restriction pattern analysis - no citations, while the methods have already been described. Also what titer is considered "high titer"? 109? 1010? Please, be more informative
Line 289 Families Myoviridae and Podoviridae are no logner existing in phage taxonomy. The names could still be used but as morphology type.
Figures 2 and 3 are very hard to read, please reconsider making the graphs bigger: bigger font, thicker lines.
Figure 4: you have icons from your programme visible on the figure.
In reviewers opinion the supplementary figures S2 and S3 should be moved to the main body of the manuscript as the data would be more clearly presented. It is hard to interpret gel without seeing it.
3.9. Phylogenetic analysis There is a lack of new nomenclature regarding the phage phylogenesis, what family do they belong to?
Lines 494-495 As the reviewer suggested Myoviridae and Podoviridae names should no longer be used regarding phage taxonomy and phylogenesis. Please use correct, actual names of the families.
Comments on the Quality of English LanguageThere are numerous typing mistakes such as V. cholera instead of V. cholerae
Grammar mistakes such as "was" instead of "where" - please read throughout the manuscript carefully and correct those mistakes.
The language is also lacking scientific soundness at times.
Author Response
Point to point response to the reviewer comment
Reviewer 2
Comment 1:
Comments and Suggestions for Authors
Line 58 O1 is in italics
Response:
We are very much thankful for your valuable comments on our manuscript.
As per the comment we have corrected O1 in line 50, which is not in italics in the revised manuscript. Inspired by your observation we have additionally we have incorporated the scientific name and ATCC number in some places in the revised manuscript highlighted in yellow.
Comment 2:
Introduction - the part about V. cholerae and it's pathogenicity and antibiotic resistance is a bit "ragged". Some sentences have no connection with one another or any logical train of thought. I.e. Lines 62-67 Do the authors mean the resistance of V. cholerae or resitance in general as the numbers suggest it refers to general problem. So how does it apply to V. cholerae? Then lines 68-74 seem to answer the question, but it also make the previous sentence redundant.
Response:
Thank you so much for the comment. This was really a deficiency in our write up as almost same concern has been raised by another reviewer. Based on the comments, we have modified the introduction part and tried to improve it. We have specifically mentioned the ongoing problem of AMR of V. cholerae and also described the importance of our study.
Comment 3:
Methods 2.1 Bacteria isolation - please name the hospitals or at least the cities in which te hospitals were localised as from this description it is impossible to know the scale of the isolation (global? local?). The names are written in the results section but it should also be pointed out in the methods. Also, what strains of Enterobacteriales were used for host range? Any ATCC reference strains or some clinical isolates? There is no mention of those strains characteristics anywhere in the manuscript
Response:
Thank you for your valuable suggestion. We have updated the section 2.1 as per your suggestion. The details are as follows:
2.1 Bacteria isolation: Clinical strains which were used in our study was collected from various hospitals in India. We have mentioned the name of those cities in the method section of our revised manuscript. Sample ID, name of the city and collection year has been mentioned in Table 1 of the revised manuscript. Moreover, ATCC number of strains used in our study for host range determination has also been mentioned in the revised manuscript.
Comment 4:
2.2 Bacteriophage isolation: again, name the cities or sewage treatment plants from which the samples were obtained, especially since you mantion that the samples from Kolkata were treated differently than the rest of the samples.
Response: Thank you for your observation and comments. We have worked on it and updated the information as per your suggestion. The details are given below
2.2 Bacteriophage isolation: Vibrio phage 4141 was isolated form clinical stool sample collected from IDBG (Infectious Disease and Beliaghata General hospital) hospital, Kolkata. Vibrio phage MJW was isolated from the sewage water sample collected from Majhdia, a city in the District of Nadia, India. We have mentioned these points in our revised manuscript.
Comment 5: Line 137 V. cholera cultures was - V. choleraE cultures WERE
Response: Thank you for your critical observation. As per the comment we have corrected Line 137. We have also corrected all other grammatical mistakes.
Comment 6:
2.4 Host range - the separation using the plaque morphology or just the titer is no logner considered informative. Instead the EOP should be used
Response: Thank you for your valuable suggestion. As suggested, we have performed the Efficiency of Plating (EOP) of both the phages with clinical strains of V. cholerae and observed that 46 clinical strains in case of 4141 phage have shown a minimum EOP >0.5. Whereas 45 samples in case of MJW phage also showed a minimum EOP >0.5., suggesting the purity and highly lytic nature of the phage. The result of the EOP has been included in the revised manuscript as table 1.
We are grateful to the reviewer for suggesting us to perform the EOP of both the phages. This suggestion has help us to improve quality of our manuscript.
Comment 7:
2.6. Phage stability (pH and Temperature) a mistake in citation number - 454?!
Response:
Thank you for your comment. As per your suggestion we have corrected the wrong citation number and new citation number 38 was incorporated in the revised manuscript,.
Comment 8:
2.9. Genomic DNA extraction and restriction pattern analysis - no citations, while the methods have already been described. Also what titer is considered "high titer"? 109? 1010? Please, be more informative
Response:
Thank you very much for indicating the mistake and helping us to improve the manuscript. As suggested, we have included the citation (Citation number 42 and 43) in method section as per the need and also mentioned the titer of the phage (1010 PFU/ml) in the revised manuscript.
Comment 9:
Line 289 Families Myoviridae and Podoviridae are no logner existing in phage taxonomy. The names could still be used but as morphology type.
Response:
We are very much thankful for your insightful comment. We have classified both the phages according to new ICTV classification and mentioned the new names in our revised manuscript.
Comment 10:
Figures 2 and 3 are very hard to read, please reconsider making the graphs bigger: bigger font, thicker lines.
Response:
Thanks for the comment. In the revised manuscript, we have improved Figure 2 and 3 with thicker lines, bigger graphs and bigger font.
Comment 11:
Figure 4: you have icons from your programme visible on the figure.
Response:
Thank you for your valuable comment. We have removed the programme icon from Figure 4 and converted the image into JPEG format. The new figure 4 is inserted in our revised manuscript.
Comment 12:
In reviewer’s opinion the supplementary figures S2 and S3 should be moved to the main body of the manuscript as the data would be more clearly presented. It is hard to interpret gel without seeing it.
Response:
Thank you very much for your valuable comment which has helped us to improve the revised manuscript. As per your suggestion, we have moved Figure S2 and Figure S3 to the main figure, and incorporated in the revised manuscript as Figure 5. Reviewer-1 also commented about the restriction endonuclease digestion pattern image that we have provided with the original manuscript; hence, we have repeated the experiment and included a new figure in the revised manuscript as figure 5A.
Comment 13:
3.9. Phylogenetic analysis There is a lack of new nomenclature regarding the phage phylogenesis, what family do they belong to?
Response:
We are thankful for your valuable comment.
As per your suggestion we have reconstructed the phylogenetic tree for both the phages with ViPTree web version 4.0 and compared the closest phage genome by tBlastX%. We have also performed phylogenetic tree construction on the basis of DNA pol I enzyme by MEGA X software and incorporated the updated figure as Figure 8, Figure 9 and Figure 10 in the revised manuscript.
Comment 14:
Lines 494-495 As the reviewer suggested Myoviridae and Podoviridae names should no longer be used regarding phage taxonomy and phylogenesis. Please use correct, actual names of the families.
Response:
Thank you very much for your valuable comments on our manuscript. We have corrected the taxonomical designation as per latest ICTV classification for bacteriophages in the revised manuscript.
Comment 15:
Comments on the Quality of English Language
There are numerous typing mistakes such as V. cholera instead of V. cholerae
Response:
Thank you so much for your attentive review and observation. As suggested, by we have gone through the manuscript, taken help from the senior colleagues who are known as good in English and accordingly corrected typing mistakes in the revised manuscript.
Comment 16:
Grammar mistakes such as "was" instead of "where" - please read throughout the manuscript carefully and correct those mistakes.
Response:
Thank you so much for your observation. As per your suggestion we have gone throughout the manuscript and corrected the grammatical mistakes.
Comment 17:
The language is also lacking scientific soundness at times.
Response:
Thanks for your thoughtful comment. We have gone through the manuscript and tried to fill out the gap to bring scientific soundness in the revised manuscript. We have taken help from the senior colleagues who are as good in English.
Finally, thanks for your time and valuable comments. We have responded to all of your comments and hope the revised manuscript may be up to the standard of publication.
Reviewer 3 Report
Comments and Suggestions for Authors
The manuscript is devoted to the isolation, characterization and genomic analysis of novel lytic vibrio-phages VC_Phage_4141 and Vibrio Phage 3 MJW isolated from sewage water samples of Kolkata.
Cholera is still a dangerous anthroponotic infection. The causative agent of cholera is Vibrio cholerae, with the advent of antibiotics, there is hope for a victory over this disease, but recently the danger of new epidemics of this infection has been increasing. Primarily due to the distribution of strains resistant to antimicrobial drugs. Undoubtedly, the development of new products of combating dangerous pathogens based on bacteriophages is an important task.
The manuscript is written according to the classical model. Isolation, purification, and titration of phages have been performed. The authors studied the stability of the phages and identified a range of sensitive hosts.
Genomic and phylogenetic analyses revealed that Vc_phage_4141 was most similar to Vibrio phage N4 and Vibrio phage Rostov 1 while Vibrio phage MJW was most similar to Vibrio phage Saratov 12 and Vibrio phage ICP2 2011-A.
In general, the manuscript is well written, the information presented is well organized. I believe that this manuscript can be published in the Viruses.
Сomments:
Lines 33, 298. they are classified under the Myoviridae and Podoviridae family of the order Caudovirales, respectively.
However, according to modern taxonomy (ICTV website https://ictv.global/taxonomy), such family and order (Myoviridae and Podoviridae family of the order Caudovirales ) do not exist. The Myo- and Podo- viral morphology of virions is usually mentioned, but these are not taxonomic ranks. These are non-taxonomic groups of bacteriophages of the Caudoviricetes class. The authors need to mention the modern viral taxonomy.
Line 286. 3. Results and Discussions:
Only the results are presented in this section. Discussions are presented in paragraph 4.
Lines 378-379. Analysis of digests on 1% agarose gel electrophoresis revealed that 4141 and MJW phages are not sensitive to BamHI, XbaI and ApaI.
I suggest removing this sentence as unimportant. What does "not sensitive" mean? Is the DNA of the phages modified or are there no corresponding restriction sites? In addition, it can be assumed from the figure that there are still separate restriction sites.
Lines 392-395. Evaluation of whole genome sequencing revealed that Vc_phage_4141contains double stranded circular genome of 38,498bp which comprises 47 open reading frames (ORFs). On the other hand, Vibrio phage MJW genome comprises a double stranded linear genome of 49,880bp which consists 64 ORFs.
The authors do not describe how they established that 4141 genome is circular and MJW genome is linear. If this is done using restriction mapping, then it needs to be described. It is known that WGS in most cases (for example, the presence of terminal repeats, redundancy during replication, etc.) represents the genomes of phages in the form of pseudo-circular structures, but this has nothing to do with the actual organization of genomes.
Lines 397-403. Further analysis revealed that Vc_phage_4141 encodes all positive CDS where ATG is the first codon in 97% ORF, however, ORF 1, ORF 18 and ORF 37 starts with the rare CGA, TTG and GTG codons. While Vibrio phage MJW sequence analysis unfolded 24 positive sense CDS and 40 reverse sense CDS which consists 95% of its first codons as ATG. First codon of ORF 1 encodes with rare codon ACG and ORF 18, ORF 29, ORF 32, ORF 53 codes with rare codon GTG.
The authors do not describe how they discovered ORFs starting from such unusual codons as CGA and ACG. Even if the authors have carefully analyzed the presence of RBS in the sequences, experimental evidence is needed for such conclusions. In their absence, it is better to limit yourself to standard codons (ATG, GTG, GTG).
Author Response
Point to point response to the reviewer comment
Reviewer 3
Thank you so much for your precious time for reviewing our manuscripts in depth. We are really thankful for your insightful comments which indeed helped us to improve the quality of our submitted manuscript. We have gone through all of your comments and addressed them sequentially. Kindly have a look below
Comment 1:
Lines 33, 298. they are classified under the Myoviridae and Podoviridae family of the order Caudovirales, respectively.
However, according to modern taxonomy (ICTV website https://ictv.global/taxonomy), such family and order (Myoviridae and Podoviridae family of the order Caudovirales ) do not exist. The Myo- and Podo- viral morphology of virions is usually mentioned, but these are not taxonomic ranks. These are non-taxonomic groups of bacteriophages of the Caudoviricetes class. The authors need to mention the modern viral taxonomy.
Response:
Thank you very much for your valuable comments on our manuscript. We have corrected the taxonomical designation as per latest ICTV classification for bacteriophages in the revised manuscript. According to the latest ICTV classification Vibrio phage 4141 and Vibrio phage MJW falls under the family Autographiviridae and Zobellviridae respectively.
Comment 2:
Line 286. 3. Results and Discussions:
Only the results are presented in this section. Discussions are presented in paragraph 4.
Response:
Thank you for indicating the mistake. Since, the Result and Discussion parts are in distinct paragraphs, as per your suggestion, we have now changed the headline of the aforementioned section and divided it into two independent sections, namely Results and Discussion in our revised manuscript.
Comment 3:
Lines 378-379. Analysis of digests on 1% agarose gel electrophoresis revealed that 4141 and MJW phages are not sensitive to BamHI, XbaI and ApaI.
I suggest removing this sentence as unimportant. What does "not sensitive" mean? Is the DNA of the phages modified or are there no corresponding restriction sites? In addition, it can be assumed from the figure that there are still separate restriction sites.
Response:
We appreciate your helpful opinion. As suggested by you we have eliminated the word “not sensitive” in the revised manuscript. Furthermore, we have repeated the experiment and incorporated the result of the restriction endonuclease digestion pattern as figure 5 in the revised manuscript.
Comment 4:
Lines 392-395. Evaluation of whole genome sequencing revealed that Vc_phage_4141contains double stranded circular genome of 38,498bp which comprises 47 open reading frames (ORFs). On the other hand, Vibrio phage MJW genome comprises a double stranded linear genome of 49,880bp which consists 64 ORFs.
The authors do not describe how they established that 4141 genome is circular and MJW genome is linear. If this is done using restriction mapping, then it needs to be described. It is known that WGS in most cases (for example, the presence of terminal repeats, redundancy during replication, etc.) represents the genomes of phages in the form of pseudo-circular structures, but this has nothing to do with the actual organization of genomes.
Response:
We sincerely appreciate your insightful comments on our manuscript.
Regarding the designation of Vc_phage_4141 as having a circular genome and Vibrio phage MJW as having a linear genome, we agree that this requires clarification. Initially, we analyzed the sequence using NCBI BLAST, which showed that most genomes similar to Vibrio phage 4141 were circular. Additionally, the joining of CDS 1 (positions 38391-38498 and 1-411) in the feature file suggested a circular configuration.
However, we further analyzed the genome using CENSOR (a tool for detecting long terminal repeats), which identified terminal repeats in the genome. These results have been incorporated into the supplementary data (Table LTR-1). Given that we did not specifically check for genome redundancy during replication, structural confirmation, or perform restriction mapping, we have now removed the circular/linear designation from the sentence to avoid potential misrepresentation.
Comment 5:
Lines 397-403. Further analysis revealed that Vc_phage_4141 encodes all positive CDS where ATG is the first codon in 97% ORF, however, ORF 1, ORF 18 and ORF 37 starts with the rare CGA, TTG and GTG codons. While Vibrio phage MJW sequence analysis unfolded 24 positive sense CDS and 40 reverse sense CDS which consists 95% of its first codons as ATG. First codon of ORF 1 encodes with rare codon ACG and ORF 18, ORF 29, ORF 32, ORF 53 codes with rare codon GTG.
The authors do not describe how they discovered ORFs starting from such unusual codons as CGA and ACG. Even if the authors have carefully analyzed the presence of RBS in the sequences, experimental evidence is needed for such conclusions. In their absence, it is better to limit yourself to standard codons (ATG, GTG, GTG).
Response: We appreciate the reviewer for important comments which will help to refine our manuscript.
We have gone through the annotation table and found that the ORFs described in this section were previously annotated with some errors; however, after reanalysing gene by gene content using NCBI Blast and Conserved domain searches, we re-annotated them appropriately, limiting ourselves to standard codons and focussing on the major aspects of the genomes. We have included the updated annotation table as Table S1 and Table S2 in the revised manuscript. Furthermore, the genome details are already mentioned in GenBank (OR233736.1 and OR248150.1).
Reviewer 4 Report
Comments and Suggestions for Authors
The widespread prevalence of antimicrobial resistance makes it important to find alternatives to antibiotics. Phage therapy has a number of advantages over antibiotics and may represent such an alternative. The manuscript by Sanjoy Biswas and colleagues presents the results of a comprehensive analysis of two newly isolated vibriophages. The authors determined the host range and lytic abilities of the phages, conducted experiments on phage stability, analyzed the restriction endonuclease digestion pattern of phage DNA, presented interesting results on studies of antibiofilm activity. The results shown in the manuscript are useful and interesting for researchers working in the field of phage therapy. Generally, the narration is consistent and well illustrated, although the manuscript needs careful proofreading. Phages are fairly well characterized from a wet biology perspective, but the genomic part requires significant improvement. Some notes:
Line 3 - Please consider the change of the title of manuscript to “Biological characterization and evaluation of the therapeutic value of Vibrio phages 4141 and MJW isolated from clinical and sewage water samples of Kolkata.”
Lines 33 and 34 - Myoviridae and Podoviridae family as well as order Caudovirales have been abolished. You should familiarize yourself with the latest ICTV classification scheme https://ictv.global/taxonomy
Line 3 and elsewhere - As far as I know, the word “vibriophage” does not need in hyphen
Lines 253-279 - Please indicate the version of software used
Line 279 - Please correct the word “VIRDIC” (should be “VIRIDIC”)
Lines 282-285 - Accession number OR248150 corresponds to a partial but not complete genome. Please explain and indicate in the text.
Line 292 - Was the capsid measured facet to facet or vertex to vertex? Please indicate.
Line 393 - The Autographiviridae Studiervirinae phages have a linear genome containing direct terminal repeats. How did you conclude that phage 4141 has a circular genome? Was it a replicative form? Did you try to find DTRs?
Line 399 - ORF 1 does not start with the CGA codon. Unfortunately, you determined the genome start point incorrectly and got some problems with numbering of ORFs and this start codon. You can try to determine DTRs or use the closest genomes to obtain a better starting point.
Section 3.8 - Perhaps you do not have to list all these ORFs and their functions if you have the annotation table. But you could highlight most interesting genes including the genes characteristic of Autographiviridae phages and adsorption apparatus.
Section 3.8 - Please indicate that the phage MJW is presented by a partial sequence.
Figures 5, 6 - The quality of the annotations is completely unacceptable. Please predict the gene function using blast and hhpred.
Figure 7 - The trees are unreadable.
Figure 8 - The 4141 genome is circularly permuted. You really should get another starting point.
Figure 9 - Colors are nice, but the labels and numbers are unreadable. Please improve the quality of this figure.
Lines 494-485 - Please classify your phages correctly according to the latest ICTV taxonomy and recommendations given in https://www.ncbi.nlm.nih.gov/pmc/articles/PMC8003253/
Author Response
Point to point response to the reviewer comment
Reviewer 4
Thank you so much for your constructive comments. We have gone through all of your comments and addressed them sequentially.Your insightful suggestions helped us to improve the quality of our work.
Comment 1:
The widespread prevalence of antimicrobial resistance makes it important to find alternatives to antibiotics. Phage therapy has a number of advantages over antibiotics and may represent such an alternative. The manuscript by Sanjoy Biswas and colleagues presents the results of a comprehensive analysis of two newly isolated vibriophages. The authors determined the host range and lytic abilities of the phages, conducted experiments on phage stability, analyzed the restriction endonuclease digestion pattern of phage DNA, presented interesting results on studies of antibiofilm activity. The results shown in the manuscript are useful and interesting for researchers working in the field of phage therapy. Generally, the narration is consistent and well illustrated, although the manuscript needs careful proofreading. Phages are fairly well characterized from a wet biology perspective, but the genomic part requires significant improvement. Some notes:
Response 1:
We are very much thankful for your valuable comments this will surely help us to refine our manuscript. As suggested, we have taken various steps to improve the genomic part such as, we have repeated various experiments and re-annotated the phage genome. As a result, we have incorporated new updated figures and tables accordingly in our revised manuscript. Hopefully, the changes made in the revised version of the manuscript will improve the genomic viewpoint.
Comment2:
Line 3 - Please consider the change of the title of manuscript to “Biological characterization and evaluation of the therapeutic value of Vibrio phages 4141 and MJW isolated from clinical and sewage water samples of Kolkata.”
Response 2:
Thank you for indicating the valuable suggestions on changing the title of our manuscript. As suggested by you, we have changed the title to “Biological characterization and evaluation of the therapeutic value of Vibrio phages 4141 and MJW isolated from clinical and sewage water samples of Kolkata.” We feel that it is now clearer and more rational. Thanks again.
Comment 4:
Lines 33 and 34 - Myoviridae and Podoviridae family as well as order Caudovirales have been abolished. You should familiarize yourself with the latest ICTV classification scheme https://ictv.global/taxonomy
Response 4:
We are very much thankful for your valuable comment about the updated ICTV classification. We have corrected the taxonomical designation as per latest ICTV classification for bacteriophages in the revised manuscript. According to modern ICTV classification the Vibrio phage 4141 and Vibrio phage MJW falls under the family Autographiviridae and Zobellviridae respectively. We have mentioned and changed the in the revised manuscript
Comment 5:
Line 3 and elsewhere - As far as I know, the word “vibriophage” does not need in hyphen.
Response 5: Thank you for your valuable comment. We have, removed the hyphen from the word “vibriophage” and renamed as vibriophage in our revised manuscript.
Comment 6:
Lines 253-279 - Please indicate the version of software used
Response 6: Thank you for suggesting to indicate the version of software. We have mentioned the version of the software in our revised manuscript.
Comment 7:
Line 279 - Please correct the word “VIRDIC” (should be “VIRIDIC”)
Response 7: Thank you for finding the mistake, this will help to refine our manuscript.
We have corrected the word to VIRIDIC in our revised manuscript.
Comment 8:
Lines 282-285 - Accession number OR248150 corresponds to a partial but not complete genome. Please explain and indicate in the text.
Response 8: We are thankful to reviewer for the valuable observation and important comments on our manuscript.
Accession number OR248150.1 (Vibrio phage MJW) corresponds to a partial genome because the genome carries partial coding sequence at each end and we indicated the partial genome in our text.
Comment 9:
Line 292 - Was the capsid measured facet to facet or vertex to vertex? Please indicate.
Response 9:
Thank you for your valuable suggestion. The measurements of both phages were analysed from vertex to vertex using Image J software version 1.5.4. We have clearly mentioned this point in our revised version of our manuscript.
Comment 10:
Line 393 - The Autographiviridae Studiervirinae phages have a linear genome containing direct terminal repeats. How did you conclude that phage 4141 has a circular genome? Was it a replicative form? Did you try to find DTRs?
Response 10:
We sincerely appreciate your insightful comments on our manuscript.
Regarding the designation of Vc_phage_4141 as having a circular genome and Vibrio phage MJW as having a linear genome, we agree that this requires clarification. Initially, we analyzed the sequence using NCBI BLAST, which showed that most genomes similar to Vibrio phage 4141 were circular. Additionally, the joining of CDS 1 (positions 38391-38498 and 1-411) in the feature file suggested a circular configuration.
However, we further analyzed the genome using CENSOR (a tool for detecting long terminal repeats), which identified terminal repeats in the genome. These results have been incorporated into the supplementary data (Table LTR-1). Given that we did not specifically check for genome redundancy during replication, structural confirmation, or perform restriction mapping, we have now removed the circular/linear designation from the sentence to avoid potential misrepresentation.
Comment 11:
Line 399 - ORF 1 does not start with the CGA codon. Unfortunately, you determined the genome start point incorrectly and got some problems with numbering of ORFs and this start codon. You can try to determine DTRs or use the closest genomes to obtain a better starting point.
Response 11:
Thank you for your valuable suggestion. We have gone through the annotation table and found that the ORFs described in this section were previously annotated with some errors; however, after reanalysing gene by gene content using NCBI Blast and Conserved domain searches, we re-annotated them appropriately, limiting ourselves to standard codons and focussing on the major aspects of the genomes. We have included the updated annotation table as Table S1 and Table S2 in the revised manuscript. Furthermore, the genome details are already mentioned in the GenBank.
Comment 12:
Section 3.8 - Perhaps you do not have to list all these ORFs and their functions if you have the annotation table. But you could highlight most interesting genes including the genes characteristic of Autographiviridae phages and adsorption apparatus.
Response 12:
Thank you for your valuable comment. As suggested by you, we have only highlighted the most interesting and important genes of the family Autographaviridae and Zobellviridae in our revised manuscript.
Comment 13:
Section 3.8 - Please indicate that the phage MJW is presented by a partial sequence.
Response 13:
Thank you for your valuable suggestion. We have indicated the MJW phage as partial genome in our amended manuscript.
Comment 14:
Figures 5, 6 - The quality of the annotations is completely unacceptable. Please predict the gene function using blast and hhpred.
Response 14:
Thank you for your valuable suggestion. I agree with you about the issue of the quality of annotation. Therefore, we have re-annotated the genome with the help of NCBI Blast database and curated the function of the gene by NCBI Blastx. Moreover, we have also analysed their conserve domain for better understand their function with the help of CD-search tool. The updated figures are incorporated in our revised manuscript as Figure 6 and Figure 7. We have used NCBI BLASTx and Conserved Domain search tool for the annotation and provided the improved and more informative figures as mentioned above in the manuscript. Hopefully, now you will find a better quality of annotation that will be acceptable for publication.
Thank you again for your critical review, this will surely help us to improve the quality of our manuscript.
Comment 15:
Figure 7 - The trees are unreadable.
Response 15:
Thank you for your valuable comments. I agree with you that the resolution of the tree was poor. We have updated the phylogenetic tree with better resolution and also, we have re-analysed the phylogenetic tree with ViPTree for better classification with improving the quality of the figure with thicker and bigger font and incorporated the updated image in our revised manuscript as figure8, figure9 and figure 10. I hope that the resolution of the new figures is better and readable that can be published.
Comment 16:
Figure 8 - The 4141 genome is circularly permuted. You really should get another starting point.
Response 16:
We are thankful for your valuable comment on our manuscript. We were not familiar about EasyFig software analysis; therefore, we have reanalysed the sequence and compared that close genomes by ViPTree genomic comparison programme. Kindly find our incorporation in the Figure 11of the revised manuscript.
Comment 17:
Figure 9 - Colors are nice, but the labels and numbers are unreadable. Please improve the quality of this figure.
Response 17:
We are thankful to the reviewer for the valuable comment. As suggested by you we have updated the figures with better resolution. Kindly find the Figure 12 in the revised manuscript
Comment 18:
Lines 494-485 - Please classify your phages correctly according to the latest ICTV taxonomy and recommendations given in https://www.ncbi.nlm.nih.gov/pmc/articles/PMC8003253/
Response 18:
We are very much thankful for your valuable comment about the updated ICTV classification. We have corrected the taxonomical designation as per latest ICTV classification for bacteriophages in the revised manuscript. According to modern ICTV classification the Vibrio phage 4141 and Vibrio phage MJW falls under the family Autographiviridae and Zobellviridae respectively. We have mentioned the changes in the revised manuscript, kindly find the citation number 61.